# Pharmacologic ATF6 activating compounds are metabolically activated to selectively modify endoplasmic reticulum proteins

Ryan Paxman[1], Lars Plate[1,2†], Erik A Blackwood[3,4], Chris Glembotski[3,4], Evan T Powers[1], R Luke Wiseman[2]*, Jeffery W Kelly[1,2,5]*

[1]Department of Chemistry, The Scripps Research Institute, La Jolla, United States; [2]Department of Molecular Medicine, The Scripps Research Institute, La Jolla, United States; [3]Department of Biology, San Diego State University, San Diego, United States; [4]San Diego State University Heart Institute, San Diego State University, San Diego, United States; [5]The Skaggs Institute for Chemical Biology, The Scripps Research Institute, La Jolla, United States

*For correspondence:
wiseman@scripps.edu (RLW);
jkelly@scripps.edu (JWK)

Present address: †Departments of Chemistry and Biological Sciences, Vanderbilt University, Nashville, United States

**Abstract** Pharmacologic arm-selective unfolded protein response (UPR) signaling pathway activation is emerging as a promising strategy to ameliorate imbalances in endoplasmic reticulum (ER) proteostasis implicated in diverse diseases. The small molecule N-(2-hydroxy-5-methylphenyl)-3-phenylpropanamide (**147**) was previously identified (*Plate et al., 2016*) to preferentially activate the ATF6 arm of the UPR, promoting protective remodeling of the ER proteostasis network. Here we show that **147**-dependent ATF6 activation requires metabolic oxidation to form an electrophile that preferentially reacts with ER proteins. Proteins covalently modified by **147** include protein disulfide isomerases (PDIs), known to regulate ATF6 activation. Genetic depletion of PDIs perturbs **147**-dependent induction of the ATF6-target gene, *BiP*, implicating covalent modifications of PDIs in the preferential activation of ATF6 afforded by treatment with **147**. Thus, **147** is a pro-drug that preferentially activates ATF6 signaling through a mechanism involving localized metabolic activation and selective covalent modification of ER resident proteins that regulate ATF6 activity.
DOI: https://doi.org/10.7554/eLife.37168.001

## Introduction

When protein misfolding or aggregation within the endoplasmic reticulum (ER) becomes excessive, cells respond to this ER stress or imbalance in protein homeostasis (or 'proteostasis') by activating the Unfolded Protein Response (UPR), a three-armed stress-responsive signaling pathway (*Hetz et al., 2015*; *Plate and Wiseman, 2017*; *Walter and Ron, 2011*). Signaling within these arms is initiated by the ER stress-sensing transmembrane proteins, ATF6, IRE1 and PERK, for which the arms are named. A key role of the UPR is to restore ER proteostasis in response to pathologic insults that induce ER stress, through a combination of transcriptional and translational mechanisms. IRE1 and ATF6 activate the stress-responsive transcription factors, spliced XBP1 (XBP1s, a spliced variant of XBP1) and ATF6 (a cleaved product of full-length ATF6), respectively, that induce the expression of overlapping, and distinct subsets of ER chaperones, folding enzymes, and degradation factors involved in maintaining ER proteostasis (*Shoulders et al., 2013*; *Lee et al., 2003*; *Yamamoto et al., 2004*). PERK activation induces a transient attenuation of new protein synthesis and activation of transcription factors, such as ATF4 that induce stress-responsive genes involved in both ER proteostasis maintenance and other aspects of cellular physiology, including redox regulation and amino

acid synthesis (*Harding et al., 1999*; *Harding et al., 2003*). Through these adaptive signaling mechanisms, the UPR functions to promote ER proteostasis and alleviate ER stress.

The capacity of the UPR signaling arms to distinctly influence ER proteostasis and function suggests that selective activation of these pathways has significant potential to alleviate pathologic imbalances in ER proteostasis associated with etiologically diverse human diseases (*Hetz et al., 2015*; *Plate and Wiseman, 2017*). In particular, activation of the ATF6 signaling arm has been shown to be useful for ameliorating disease-associated imbalances in ER proteostasis and function. We previously showed that stress-independent activation of the ATF6 transcription factor using a chemical genetic approach induces protective remodeling of ER proteostasis pathways to selectively reduce secretion and extracellular aggregation of destabilized, amyloid disease-associated proteins, such as transthyretin and immunoglobulin light chain, without significantly impacting the secretion of the endogenous proteome (*Shoulders et al., 2013*; *Chen et al., 2014*; *Cooley et al., 2014*; *Plate et al., 2016*). Overexpression of the active ATF6 transcription factor in the heart also has been shown to improve cardiac performance in mouse models of ischemic heart disease, through a mechanism involving ATF6-dependent regulation of the antioxidant gene, catalase (*Jin et al., 2017*). Similarly, overexpression of the active ATF6 transcription factor in the liver improves insulin sensitivity in obese mice (*Ozcan et al., 2016*). These results indicate that ATF6 activation offers a unique therapeutic opportunity to ameliorate ER proteostasis defects implicated in diverse diseases.

The potential for ATF6 activation as a therapeutic approach is further demonstrated by human genetic studies. These show that patients harboring *ATF6α* mutations that render this pathway constitutively active present with the retinal development disorder, achromatopsia, but notably do not show defects in other organs or systemic tissues resulting from constitutive ATF6 activity (*Chiang et al., 2017*; *University of Washington Center for Mendelian Genomics et al., 2015*; *Kohl et al., 2015*). This indicates that pharmacologic activation of ATF6 in adults is also likely to be well tolerated and not lead to systemic defects. The therapeutic potential for ATF6 activation demonstrated by these genetic results combined with the potential to correct imbalances in ER proteostasis associated with diverse diseases through ATF6 activation has led to a significant effort to identify small molecule pharmacologic activators of ATF6 that similarly correct pathologic imbalances in ER proteostasis implicated in disease.

We recently reported the discovery of several small molecules that preferentially activate the ATF6 arm of the UPR, which were identified using a cell-based-reporter high-throughput screening (HTS) strategy (*Plate et al., 2016*). One of the more promising of these compounds, *N*-(2-hydroxy-5-methylphenyl)-3-phenylpropanamide (**147**), was further characterized by whole cell RNAseq and proteomic profiling to confirm preferential activation of the ATF6 transcriptional program. Importantly, **147** phenocopied protective benefits associated with genetic ATF6 activation. For example, treatment with **147** reduced the secretion and extracellular aggregation of destabilized, amyloidogenic variants of transthyretin and immunoglobulin light chain in amyloid disease relevant cell models (*Plate et al., 2016*). And notably, **147** did not influence secretion of the endogenous secretory proteome. Furthermore, **147**-dependent ATF6 activation was recently shown to promote stem cell differentiation of embryonic stem cells to mesodermal lineages, revealing a previously unknown role for ATF6 in dictating stem cell specification (*Kroeger et al., 2018*). Thus, **147** has emerged as a promising compound for further development as a pharmacologic ATF6 activator that can be employed to enhance ER proteostasis in the context of human disease.

Despite this promise, the mechanism of how **147** preferentially activates the ATF6 arm of the UPR is unclear. In response to ER stress, ATF6 is activated through a process involving reduction of inter- and intra-molecular disulfides that results in formation of a reduced ATF6 monomer that is trafficking competent (*Nadanaka et al., 2007*; *Nadanaka et al., 2006*). This reduced monomer then traffics to the Golgi where it is processed by the Site 1 and Site 2 proteases (S1P and S2P, respectively), releasing the N-terminal, active ATF6 transcription factor responsible for upregulation of the ATF6 transcriptional program (*Ye et al., 2000*). Previously, we showed that treatment with **147** activates endogenous ATF6 through a similar mechanism involving increased processing by S1P and S2P to release the active ATF6 transcription factor (*Plate et al., 2016*). However, the protein target(s) and molecular mechanism(s) responsible for the **147**-dependent increases in ATF6 activation are

currently unknown. The ATF6 protein has no known small molecule binding sites that could serve to allosterically activate this protein, so it seems likely that **147** initiates signaling by interacting with proteins critical for regulating ATF6 activity. However, this hypothesis remains untested.

Here, we report our efforts focused on defining the molecular mechanism of **147**-dependent ATF6 activation. We find that the acylated 2-amino-*p*-cresol substructure of **147**, which is shared by several other pharmacologic ATF6 activators identified in our HTS screen (*Plate et al., 2016*), is required for pharmacologic ATF6 activation by this compound. Given that chemicals comprising *p*-alkyl phenol substructures are oxidized in cells by cytochrome P450s to yield highly reactive *p*-quinone methides (*Bolton, 2014*), the dependence of **147**-dependent ATF6 activation on the 2-amino-*p*-cresol moiety suggested that metabolic oxidation could be involved in the activity of **147**. We find that **147** is metabolically activated in a process sensitive to the P450 inhibitor, resveratrol. This activation results in the formation of an electrophile that reacts with multiple proteins to form covalent conjugates. A key observation is that the proteins modified by **147** are primarily localized to the ER, which is not a general property of reactive electrophiles (*Trott et al., 2008*), and is consistent with the metabolic activation of this compound by an ER-localized cytochrome P450(s). Interestingly, ER protein disulfide isomerases (PDIs), a class of proteins involved in regulating and catalyzing disulfide formation and breakage within the ER, are enriched among the proteins modified by **147**. PDIs have been previously implicated in regulating the formation and dissolution of ATF6 disulfides critical for initiating ATF6 trafficking and subsequent ATF6 transcription factor activation during ER stress (*Higa et al., 2014*). This suggests that **147**-dependent ATF6 activation is mediated by targeting PDI activity. Consistent with this notion, shRNA-mediated knockdown of select PDIs impairs **147**-dependent induction of the ATF6-target gene, *BiP*. Our results indicate that **147**-dependent ATF6 activation proceeds through a mechanism requiring metabolic activation close to or within the ER to produce a reactive electrophile that selectively modifies a subset of ER proteins, including multiple PDIs, that enhances ATF6 activity by increasing the ER population of the reduced, trafficking-competent ATF6 monomer.

## Results

### The 2-amino-p-cresol substructure of 147 is necessary for compound-dependent ATF6 activation

Compound **147** consists of a 2-amino-*p*-cresol substructure (i.e. the A-ring) connected to a second aromatic ring (the B-ring) via a carbon-based chain (the linker; see *Figure 1A*). We prepared analogs of **147** where the A-ring was varied by converting 3-phenylpropanoic acid to 3-phenylpropionyl chloride using oxalyl chloride and then coupling it to a variety of aromatic amines (*Figure 1—figure supplement 1A*). The activities of these compounds were determined by measuring their ability to activate a previously described ATF6 transcriptional reporter that consists of a fragment of the ATF6-regulated *BiP* promoter driving expression of firefly luciferase (referred to as ERSE.FLuc) in HEK293T cells (*Plate et al., 2016*; *Yoshida et al., 1998*). Interestingly, removal of the phenolic hydroxyl group in the A-ring (**147-1**) or replacement of it with a fluoro group (**147-2**) yields compounds that do not activate this ATF6 reporter (*Figure 1B*). Compounds where the methyl group of **147** is removed (**147-3**) or where the methyl is replaced with a trifluoromethyl group (**147-4**) also do not activate this reporter. Compound **147–4** also does not induce ATF6 target genes such as *BiP* in human embryonic stem cells (*Kroeger et al., 2018*). Furthermore, moving the hydroxyl group to the *ortho* or *meta* positions of the cresol moiety (**147–5** and **147–6**) also results in compounds that do not activate the ATF6 reporter. These results indicate that the 2-amino-*p*-cresol substructure of **147**, in its entirety, is required for compound-dependent ATF6 activation.

We next explored the dependence of **147** activity on the linker connecting the A- and B-rings (*Figure 1C*). Analogs of **147** in which the linker length or structure was varied were prepared by coupling 2-amino-*p*-cresol to benzoic acid (**147-7**), phenylacetic acid (**147-8**), 4-phenylbutanoic acid (**147-9**) and cinnamic acid (**147-10**) via their respective acyl chlorides (*Figure 1—figure supplement 1B*). Interestingly, all of these analogs showed diminished capacity to activate the ATF6 reporter relative to **147**, indicating that the length and flexibility of the 3-carbon linker are critical for the activity of **147** (*Figure 1C*).

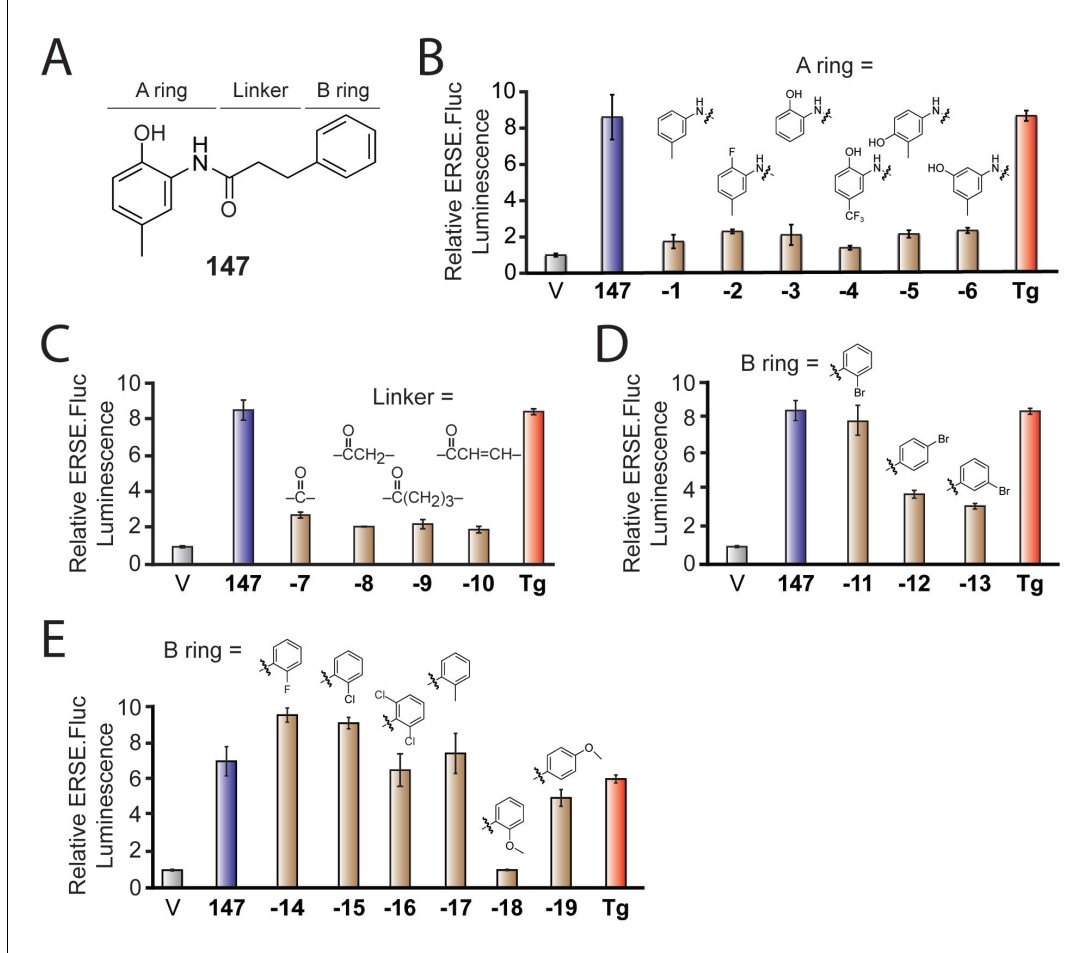

**Figure 1.** Activities of **147** analogs toward ATF6 activation. (**A**) Structure of **147** with its three components – the A-ring, B-ring, and linker – indicated. (**B**) Bar graph showing the relative activation of the ERSE.FLuc ATF6 transcriptional reporter in HEK293T cells treated with **147** (10 µM; 18 hr), the ER stressor thapsigargin (Tg; 0.5 µM; 18 hr), or the indicated analogs of **147** where the A-ring was varied (10 µM; 18 hr). Error bars show SEM for three technical replicates. (**C**) Bar graph as in (**B**) where HEK293T cells were treated with the indicated **147** analogs where the linker was varied (10 µM; 18 hr). Error bars show SEM for three technical replicates. (**D**) Bar graph as in (**B**) where HEK293T cells were treated with the indicated **147** analogs in which a bromine atom was placed in the *ortho*, *para*, and *meta* positions of the B-ring (10 µM; 18 hr). Error bars show SEM for three technical replicates. (**E**) Bar graph as in (**B**) where HEK293T cells were treated with the indicated **147** analogs with a variety of substituents incorporated onto the B-ring. Error bars show SEM for three technical replicates.

DOI: https://doi.org/10.7554/eLife.37168.002

The following figure supplement is available for figure 1:

**Figure supplement 1.** Synthetic schemes used to prepare analogs of **147**.
DOI: https://doi.org/10.7554/eLife.37168.003

Finally, we explored how substitutions on the B-ring influence **147**-dependent ATF6 activation. Analogs of **147** with B-ring substituents were prepared by coupling 2-amino-*p*-cresol to appropriately substituted 3-phenylpropanoic acids (*Figure 1—figure supplement 1C*). Incorporating a single bromine atom at different positions around the B-ring demonstrated that the *ortho* position is amenable to substitution, as the *o*-Br analog (**147-11**) showed activation of the ATF6 reporter to levels comparable to the parent compound (*Figure 1D*). In contrast, the *p*-Br (**147-12**) or *m*-Br (**147-13**) analogs showed lower levels of reporter activation. Fluorine, chlorine, and methyl substituents at a single *ortho* position or Cl atoms at both *ortho* positions were also tolerated (*o*-F: **147–14**; *o*-Cl: **147–15**; *o*-Cl$_2$: **147–16**; *o*-CH$_3$: **147–17**; *Figure 1E*). Methoxy groups were tolerated at the *para* position of the B ring, but not the *ortho*, position (*o*-OCH$_3$: **147–18**; *p*-OCH$_3$: **147–19**).

These medicinal chemistry efforts demonstrate that the 2-amino-*p*-cresol A ring is required for **147**-dependent ATF6 activation, whereas the B-ring is more tolerant of structural perturbation. Interestingly, we also found that the length and flexibility of the linker is important for the activity of **147**, likely reflecting the need for a specific conformation and relative positioning of the two rings for **147**-dependent ATF6 activation.

## 147-dependent ATF6 activation is inhibited by exogenous thiols and a cytochrome P450 inhibitor

The requirement for the 2-amino-*p*-cresol substructure for ATF6 activation indicates that this component is central to the activity of **147**. Interestingly, some compounds with similar substructures are oxidized to highly protein reactive structures in cells and in vivo (*Bolton, 2014*). Eugenol (4-allyl-2-methoxyphenol; *Figure 2—figure supplement 1A*) is a particularly well-studied example of a *p*-alkyl phenol that is oxidized to a *p*-quinone methide by cytochrome P450s in cells (*Bolton et al., 1995*; *Thompson et al., 1991*; *Thompson et al., 1995a*; *Thompson et al., 1995b*; *Thompson et al., 1993*). This *p*-quinone methide (*Figure 2—figure supplement 1A*) can react with nucleophiles resulting in depletion of glutathione and modification of cellular proteins, presumably through reactions with thiol side chains of reactive cysteines (*Thompson et al., 1991*). In addition, *p*-cresol itself can be oxidized to a *p*-quinone methide in rat and human microsomes in a process that is catalyzed by cytochrome P450s (*Thompson et al., 1995b*; *Yan et al., 2005*). These observations, combined with the requirement of the 2-amino-*p*-cresol substructure for **147** activity, suggest that **147**-dependent ATF6 activation could similarly involve oxidation to the *p*-quinone methide (**147-QM**) shown in *Figure 2A*.

Alternatively, metabolic activation of **147** could occur by oxidation of the 2-acylated amino *p*-cresol to an *o*-quinone imine (**147-QI**; *Figure 2A*). This process is directly analogous to the well-known oxidation of the analgesic antipyretic acetaminophen (*Figure 2—figure supplement 1B*) to N-acetyl-*p*-benzoquinone imine (*Dahlin et al., 1984*). The **147-QI** intermediate can spontaneously tautomerize to the reactive *p*-quinone methide **147-QM**, which could initiate ATF6 activation through the covalent modification of proteins (*Figure 2A*) (*Bolton, 2014*). The cellular components most likely responsible for initiating this oxidation mechanism are cytochrome P450s, which are generally responsible for metabolizing xenobiotic compounds, although other phase 1 enzymes are also capable of catalyzing the necessary oxidation steps (*Bolton, 2014*; *Cao and Peng, 2014*).

We initially scrutinized the metabolic activation mechanism for **147** shown in *Figure 2A* by monitoring activation of the ERSE.FLuc ATF6 reporter in HEK293T cells pre-treated with the free thiol-containing compound β-mercaptoethanol (BME). If **147** activity requires oxidation to an electrophile that covalently modifies Cys residues of proteins involved in ATF6 regulation, the exogenous addition of BME should disrupt this mechanism either by directly reacting with the quinone methide or by increasing the reducing potential in the cell so free thiols on a competing molecule (e.g. glutathione) react with **147-QM** before it can react with its protein targets. Consistent with this notion, increasing doses of BME significantly reduced **147**-dependent activation of the ATF6 reporter (*Figure 2B*), but did not influence activation of the ATF6 reporter afforded by treatment with the global ER stressor, thapsigargin (Tg; *Figure 2C*). Similar results were observed upon co-treatment with the alternative free thiol containing compound, N-acetyl cysteine (NAC; *Figure 2—figure supplement 1C*). Collectively, these results show that increases in intracellular free thiols reduce **147**-dependent ATF6 activation.

Next, we probed the dependence of ATF6 activation by **147** on cytochrome P450s by assessing activation in the presence of a P450 inhibitor. We co-treated cells with **147** and resveratrol – a compound that is both an anti-oxidant and an inhibitor of CYP1A1, CYP1A2, and CYP1B1 (*Chang et al., 2001*). Resveratrol alone modestly increased basal activation of the ERSE.Fluc reporter, indicating that this compound either mildly activates ATF6 or stabilizes the firefly luciferase reporter (*Figure 2D*). In contrast, co-treatment with resveratrol and **147** significantly reduced the **147**-dependent increase in ATF6 reporter activation (*Figure 2D*). Notably, co-treatment with resveratrol and Tg did not reduce Tg-dependent activation of the ATF6 reporter (*Figure 2E*). The reduction in **147**-dependent ATF6 activation afforded by resveratrol co-treatment is consistent with **147** activity involving P450-dependent oxidation to a reactive electrophile(s), as indicated in *Figure 2A*.

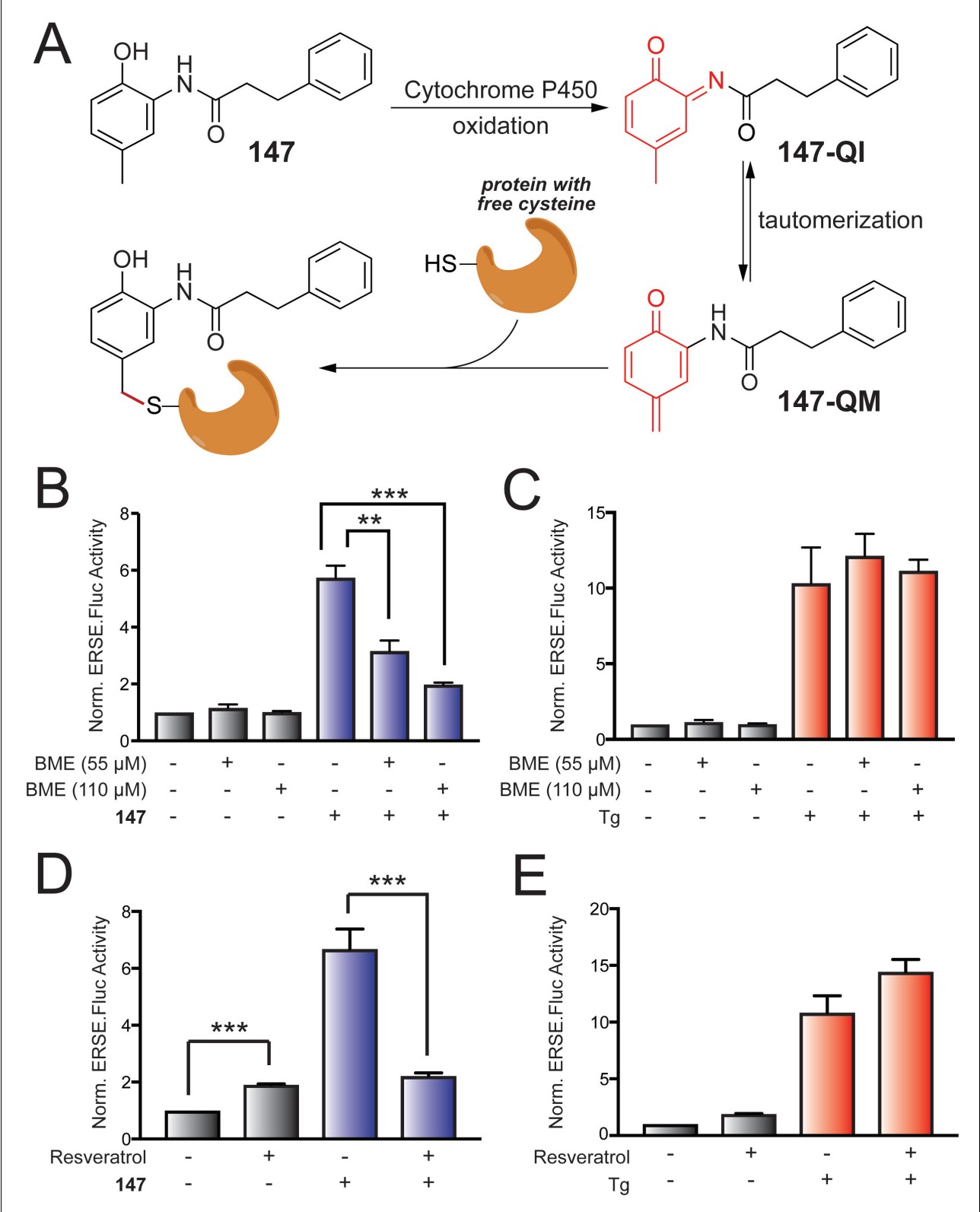

**Figure 2.** 147-dependent ATF6 activation is sensitive to increases in cellular free thiols and resveratrol. (**A**) Illustration showing a mechanism whereby 147 is converted to an *o*-quinone imine (**147-QI**) followed by tautomerization to a quinone methide (**147-QM**), which can then react with cellular proteins through nucleophiles such as free Cys side chains. (**B**) Bar graph showing the activation of the ERSE.FLuc ATF6 reporter in HEK293T cells treated with 147 (10 µM) and/or β-mercaptoethanol (BME; 55 µM or 110 µM) for 18 hr. Error bars show SEM for 3 independent experiments. *p<0.05.
*Figure 2 continued on next page*

Figure 2 continued

(C) Bar graph showing the activation of the ERSE.FLuc ATF6 reporter in HEK293T cells treated with thapsigargin (Tg; 0.5 µM) and/or β-mercaptoethanol (BME; 55 µM or 110 µM) for 18 hr. Error bars show SEM for 3 independent experiments. (D) Bar graph showing the activation of the ERSE.FLuc ATF6 reporter in HEK293T cells treated with 147 (10 µM) and/or resveratrol (10 µM) for 18 hr. Error bars show SEM for 4 independent experiments. ***p<0.005. (E) Bar graph showing the activation of the ERSE.FLuc ATF6 reporter in HEK293T cells treated with Tg (500 nM) and/or resveratrol (10 µM) for 18 hr. Error bars show SEM for 4 independent experiments.

DOI: https://doi.org/10.7554/eLife.37168.004

The following figure supplement is available for figure 2:

**Figure supplement 1.** 147-dependent ATF6 activation is sensitive to increases in cellular free thiols and resveratrol.

DOI: https://doi.org/10.7554/eLife.37168.005

## The oxidation of 147 results in the covalent modification of cellular proteins

The above results support the model shown in *Figure 2A* whereby **147** is metabolically activated by cytochrome P450s to produce a reactive electrophile such as a quinone methide (**147-QM**), either directly, or through the intermediacy of **147-QI**. The **147-QM** could covalently modify proteins important for regulating ATF6 activation. We further tested this prediction in cell culture models using affinity purification mass spectrometry experiments.

We installed an alkyne substructure onto **147** by converting the Br of the methoxymethyl-protected variant of **147–11** to an alkyne (**147-20a**) using a Songashira coupling with trimethylsilyl acetylene followed by deprotection and desilylation (*Figure 3—figure supplement 1A*) (*Hundertmark et al., 2000*). The resulting alkyne-substituted **147** analog (**147-20**) had activity comparable to **147** in our ATF6 reporter assay, demonstrating that the inclusion of the alkyne did not diminish the compound's ability to activate ATF6 (*Figure 3A*). The alkyne in **147–20** allows for a copper-catalyzed azide-alkyne cyclization (CuAAC) 'click' reaction (*Besanceney-Webler et al., 2011*) with an azide-functionalized biotin (*Yang et al., 2010*), which can be used as a handle to purify proteins covalently modified by **147–20** through the same mechanism shown in *Figure 2A* (*Figure 3B*). A cleavable diazo linker allows **147–20**-protein conjugates to be cleaved from the affinity column.

In order to identify proteins covalently modified by **147–20**, we prepared lysates from ALMC-2 cells treated with **147–20** for 18 hr. ALMC-2 cells are a plasma cell line derived from a light chain amyloidosis patient, in which it has been shown that **147** activates the ATF6 arm of the UPR (*Plate et al., 2016*; *Arendt et al., 2008*). These lysates were treated with an excess of biotin–diazo-linker–azide (*Figure 3B*) under 'click' chemistry conditions, enabling proteins covalently modified by **147–20** to be affinity isolated using streptavidin beads. Numerous **147–20**-based protein conjugates isolated by affinity chromatography were identified by SDS-PAGE (*Figure 3C*). Co-treatment of cells with **147–20** together with a 5-fold excess of the parent compound **147** effectively eliminated these bands, indicating that **147** and **147–20** are reacting with the same subset of proteins. Notably, co-treatment with a 5-fold excess of the inactive analog **147–4** (*Figure 1B*), having a trifluoromethyl group on the A-ring instead of a methyl group, did not block **147–20** labeling, suggesting that conjugate formation with a subset of proteins is what is activating the ATF6 arm of the UPR. As expected, co-treatment with BME or resveratrol reduced **147–20** conjugate formation, further confirming the sensitivity of the oxidative ATF6 activation mechanism to these compounds (*Figure 3—figure supplement 1B*). In gel digestion of the **147–20** conjugates shown in *Figure 3C* followed by mass spectrometry characterization identified four ER protein disulfide isomerases (PDIA1, PDIA4, PDIA6, and TXNDC5), a proteasome subunit (PSME2), and a chloride channel (CLIC1) as the most prevalent bands appearing in *Figure 3C* that were efficiently eliminated by adding a 5-fold excess of **147**. Interestingly, we did not observe covalent modification of a FLAG-tagged ATF6 in HEK293T cells treated with **147** or **147–20** or the combination of **147** and **147–20**, indicating that **147** does not appear to directly modify ATF6 (*Figure 3—figure supplement 1C*). Instead, these results suggest that **147**-dependent ATF6 activation involves **147**-mediated conjugate formation with a few select proteins that regulate ATF6 activity.

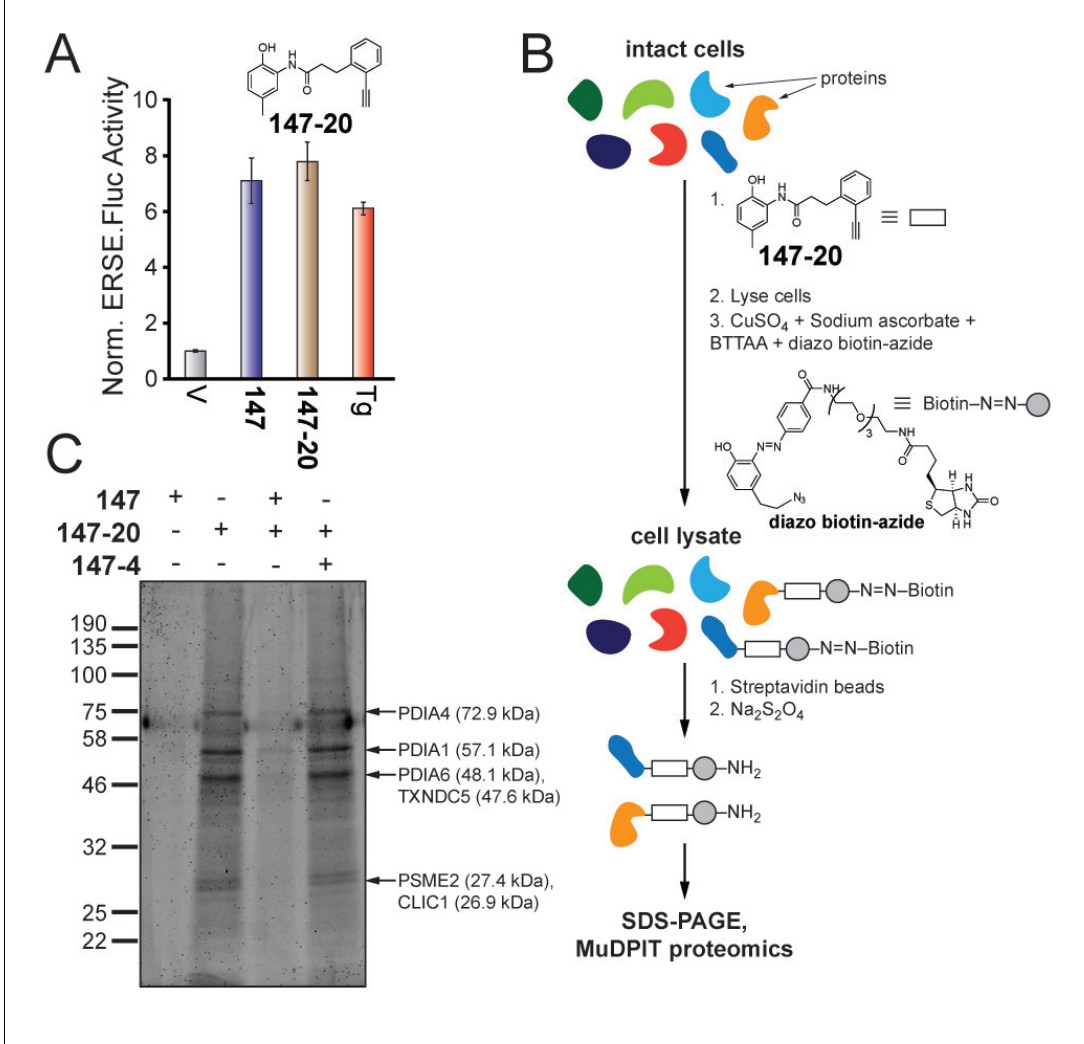

**Figure 3.** Compound **147** covalently modifies cellular proteins. (**A**) Bar graph showing the relative activation of the ERSE.FLuc ATF6 reporter in HEK293T cells treated with **147** (10 μM), **147–20** (10 μM), or thapsigargin (Tg; 0.5 μM) for 18 hr. Error bars show SEM for three technical replicates. (**B**) Schematic showing the protocol for affinity purification of proteins covalently modified by **147–20**. (**C**) Coomassie-stained SDS-PAGE of affinity purified proteins from ALMC-2 cells treated with **147** (10 μM), or **147–20** (10 μM), or the combination of the **147–20** (10 μM) and **147** (50 μM), or **147–20** (10 μM) and **147–4** (50 μM) in combination for 18 hr. Specific proteins identified by mass spectrometry of excised bands are indicated.

DOI: https://doi.org/10.7554/eLife.37168.006

The following figure supplement is available for figure 3:

**Figure supplement 1.** Compound **147** covalently modifies cellular proteins.

DOI: https://doi.org/10.7554/eLife.37168.007

## 147 preferentially modifies ER-localized proteins

To more comprehensively identify the cellular proteins forming covalent conjugates with **147–20**, we performed an affinity purification–mass-spectrometry-based proteomics analysis in three distinct cell types treated with **147–20** using the work flow outlined in *Figure 3B*. We individually treated ALMC-2, HEK293T, and liver-derived HepG2 cells with **147** (10 μM), or **147–20** (10 μM), or **147–20** (10 μM) in combination with **147** (50 μM) for 18 hr (**147** was previously shown to activate ATF6 in all three cell lines employed [*Plate et al., 2016*]). The lysates were collected, treated with the diazo biotin-azide linker under 'click' chemistry conditions, and the **147–20**-protein conjugates were affinity purified as described above, cleaved with 50 mM sodium dithionite from the affinity resin, and eluted in 1% SDS. The eluted protein conjugates were subjected to Tandem Mass Tag (TMT)-Multidimensional Protein Identification Technology (MuDPIT) analysis, as we have described previously

(*Mortenson et al., 2018*). Several hundred proteins were identified in each cell line. However, only 90 proteins were identified in all eight replicates (two biological sets of technical duplicates in ALMC-2 cells and one set of technical duplicates each in HEK293T and HepG2 cells). Of these 90 proteins, 19 exhibited significant depletion in cells incubated with 5-fold excess of **147**, indicating specific labeling. Hereafter, we define the competition TMT reporter ion ratio as the ratio between the samples treated with **147–20** to those co-treated with **147–20** and a 5-fold excess of **147**. Significant depletion is defined as a competition TMT reporter ion ratio that is significantly greater than 1.0 with a one-tailed Wilcoxon signed-rank test p value < 0.05. These 19 proteins are listed in *Table 1*. Their median competition TMT reporter ion ratios vary from 1.4 to 2.8. The full list of 90 proteins is available in *Table 1—source data 1* and the list of all identified proteins is available in *Table 1—source data 2*. Notably, ATF6 itself was not identified in any of the cell types, consistent with our results showing **147–20** does not appear to covalently modify ATF6 (*Figure 3—figure supplement 1B*).

**Table 1.** Protein targets that are covalently modified by **147**.

| Protein name | Symbol | Uniprot accession | Competition Ratio[*] | Redox?[†] | ER-localization?[‡] |
|---|---|---|---|---|---|
| Cytoskeleton-associated protein 4 | CKAP4 | Q07065 | 2.8 (1.7, 3.5) | N | Y |
| Protein disulfide isomerase A4 | PDIA4 | P13667 | 2.4 (1.8, 3.3) | Y | Y |
| Protein disulfide isomerase A6 | PDIA6 | Q15084 | 2.4 (2.0, 3.0) | Y | Y |
| Protein disulfide isomerase A1 (a.k.a. Prolyl 4-hydroxylase subunit beta) | PDIA1 (a.k.a. P4HB) | P07237 | 2.3 (2.1, 3.1) | Y | Y |
| Protein disulfide isomerase A3 | PDIA3 | P30101 | 2.2 (1.8, 3.1) | Y | Y |
| Thioredoxin | TXN | P10599 | 2.2 (1.2, 3.5) | Y | N |
| Thioredoxin domain-containing protein 5 | TXNDC5 | Q8NBS9 | 2.2 (2.0, 2.8) | Y | Y |
| Heme oxygenase 2 | HMOX2 | P30519 | 2.2 (1.4, 4.3) | N | Y |
| Chloride intracellular channel 1 | CLIC1 | O00299 | 1.9 (1.5, 2.3) | N | N |
| Thioredoxin-related transmembrane protein 1 | TMX1 | Q9H3N1 | 1.9 (1.5, 4.3) | Y | Y |
| Voltage dependent anion channel 2 | VDAC2 | P45880 | 1.6 (1.2, 1.9) | N | N[§] |
| Poly(rC)-binding protein 1 | PCBP1 | Q15365 | 1.6 (1.4, 1.8) | N | N |
| Ribosome-binding protein 1 | RRBP1 | Q9P2E9 | 1.5 (1.3, 1.8) | N | Y |
| Proteasome subunit alpha type 1 | PSMA1 | P25786 | 1.5 (1.1, 1.7) | N | N |
| D-3-phosphoglycerate kinase | PHGDH | O43175 | 1.5 (1.4, 1.9) | N | N |
| Histocompatibility minor 13 | HM13 | Q8TCT9 | 1.5 (1.2, 2.0) | N | Y |
| Polyadenylate-binding protein 4 | PABPC4 | Q13310 | 1.5 (1.1, 1.7) | N | N |
| Ribophorin 1 | RPN1 | P04843 | 1.4 (1.3, 1.8) | N | Y |
| ER resident protein 29 | ERP29 | P30040 | 1.4 (1.0, 1.8) | N | Y |

[*]Competition ratio = Ratio of TMT reporter ions for the protein in question between samples treated with **147–20** and those co-treated with **147–20** and **147**. The median value across all cell lines and replicates are shown, with the values at the 1st and 3rd quartiles shown in parenthesis.

[†]Indicates whether the protein in question has the 'cell redox homeostasis' GO annotation (GO:0045454).

[‡]Indicates whether the protein in question has the 'endoplasmic reticulum part' GO annotation (GO:0044432).

[§]VDAC2 is a mitochondrial membrane protein but has been shown to localize to sites of mitochondria that are associated with the ER.

DOI: https://doi.org/10.7554/eLife.37168.008

The following source data is available for Table 1:

**Source data 1.** Excel spreadsheet showing the competition ratio data for all 90 proteins that were identified as targets of **147** in every replicate and every cell type.

DOI: https://doi.org/10.7554/eLife.37168.009

**Source data 2.** Excel spreadsheet showing all the proteins identified among the affinity-purified targets of **147**.

DOI: https://doi.org/10.7554/eLife.37168.010

It is immediately apparent that more than a third of the 19 identified proteins are involved in redox homeostasis, including six protein disulfide isomerases (PDIA1, PDIA3, PDIA4, PDIA6, TMX1 and TXNDC5) and thioredoxin (TXN). This observation can be quantified using the PANTHER Over-representation Test to identify significantly overrepresented Gene Ontology (GO) terms from the 'biological process' domain (*Mi et al., 2017*). We find that the seven redox proteins noted above are all annotated with the GO term 'cell redox homeostasis' (GO:0045454), corresponding to a > 100 fold enrichment (False Discovery Rate (FDR) = $6.6 \times 10^{-9}$) relative to the expected number of occurrences of this annotation in a random sample of 19 genes. The cysteine residues in proteins of this class can react with electrophiles like the *p*-quinone methide predicted to be formed upon metabolic oxidation of **147** and **147–20** (*Figure 2A*). Other enriched GO terms include 'response to ER stress' (GO:0034976; 31-fold enriched, FDR = $7.9 \times 10^{-6}$) and 'protein folding' (GO:0006457; 33-fold enriched, FDR = $8.2 \times 10^{-6}$). In both cases, the enrichment is largely due to the presence of the protein disulfide isomerases (PDIs) in the list of 19 proteins identified in our TMT-MuDPIT proteomic analysis.

It is worth noting that many of the proteins in the list not involved in cellular redox homeostasis are nevertheless known to be regulated by the oxidation states of reactive cysteine residues. HMOX2 (heme oxygenase 2), unlike its paralog HMOX1 (which we did not find to be labeled by oxidized **147**), has two Cys-Pro sequences near its C-terminus. When these cysteines are oxidized, the C-terminus is unstructured and HMOX2 binds a single heme in its catalytic site (*Bagai et al., 2015*; *Fleischhacker et al., 2015*). However, this region becomes helical when the cysteines are reduced to allow binding of a second heme (*Bagai et al., 2015*; *Fleischhacker et al., 2015*). Similarly, CLIC1 adopts two substantially different folded structures depending on the oxidation state of two of its cysteine residues (*Littler et al., 2004*). When the cysteine residues are reduced, it is a soluble monomer with a structure homologous to glutathione S-transferase. When the cysteines are oxidized to form a disulfide bond, CLIC1 undergoes a drastic structural transition, becoming a helical dimer that can insert into membranes and form chloride ion channels (*Littler et al., 2004*; *Goodchild et al., 2009*).

Twelve of the nineteen proteins (63%) in the target list are annotated as ER proteins (GO:0005783, 7.4-fold enriched, FDR = $2.0 \times 10^{-6}$), demonstrating that **147–20** preferentially reacts with ER-localized proteins. These include the six PDIs noted above, along with HMOX2, HM13 (histocompatibility minor 13, also known as signal peptide peptidase or SPP; a peptidase that degrades signal peptides after they have been released from pre-proteins [*Voss et al., 2013*]), CKAP4 (cytoskeleton associated protein 4, also known as CLIMP-63; a transmembrane protein involved in the regulation of ER morphology [*Sandoz and van der Goot, 2015*]), RRBP1 (ribosome binding protein 1; a ribosome-binding protein on the rough ER [*Savitz and Meyer, 1990*]), RPN1 (ribophorin 1; a component of oligosaccharyl transferase, an enzyme complex that catalyzes protein N-glycosylation [*Wilson et al., 2008*]), and ERP29 (ER protein 29; a structural homolog of PDIs that lacks an active thioredoxin domain but retains chaperone activity [*Mkrtchian and Sandalova, 2006*; *Suaud et al., 2011*]). In addition, while VDAC2 is a mitochondrial protein, it has been shown to localize to sites of ER-mitochondrial contacts called mitochondrial-associated ER membranes or MAMs (*Naghdi and Hajnóczky, 2016*).

It is very important to note that the remarkable preference of **147–20** to react with ER-localized proteins (63% of the conjugates formed are with ER proteins) is not a general feature of other small molecule electrophiles (*Figure 4A*). Backus et al. examined the reactivity of a variety of fragment electrophiles containing cysteine-reactive chloroacetamide or acrylamide groups in live cells (*Backus et al., 2016*). Among the fragment electrophiles that modified 10 or more proteins with some degree of selectivity (as defined by Backus et al. [*Backus et al., 2016*]), proteins annotated with the GO term for ER localization (GO:0005783) generally constituted ≈ 10% of the targets that they modified (*Figure 4A*). This is comparable to the proportion of ER-localized proteins for the human proteome as a whole (8.6%). At most, ER-localized proteins made up 30% (4 out of 13) of the targets for compound **B-51** (*Figure 4A*), but this still does not represent a significant enrichment after multiple-testing correction using the PANTHER Overrepresentation Test.

Moreover, modification of ER-localized proteins and subsequent activation of UPR signaling pathways is also not a general feature of compounds known to form quinone methides. For example, celastrol—a triterpenoid natural product that harbors a quinone methide—activates many stress-responsive signaling pathways, including the heat shock response and the antioxidant response,

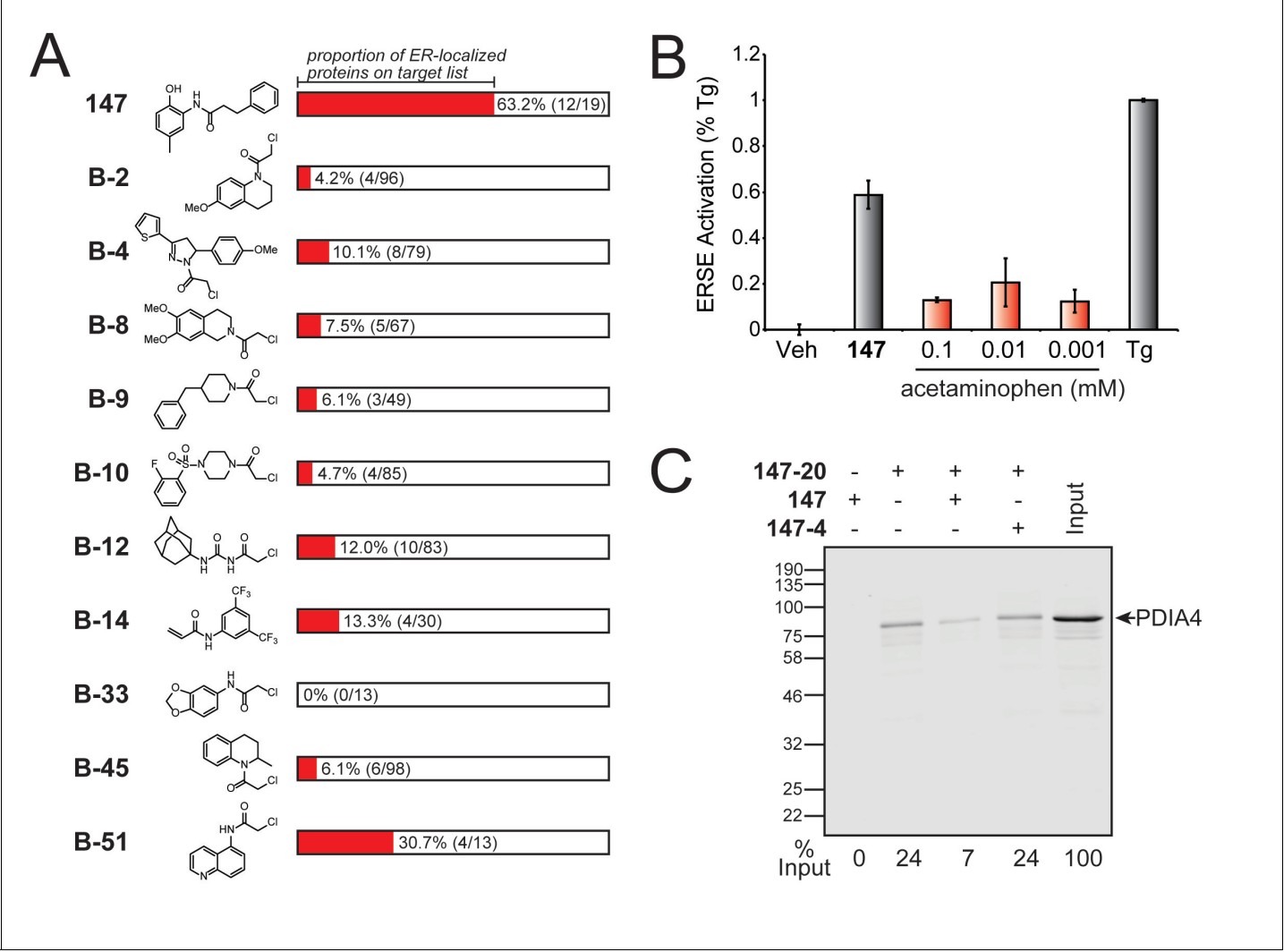

**Figure 4.** Comparison of S-reactive electrophiles demonstrating that compound **147** selectively modifies ER-localized proteins. (**A**) Proportion of ER-localized proteins covalently labeled by **147** or fragment electrophiles reported in (*Backus et al., 2016*). The ER-localized proportion for each electrophile indicated in red. The numbers in parentheses indicate the number of ER-localized proteins in a given electrophile's target list on the left and the total number of proteins in the target list on the right. The electrophiles from (*Backus et al., 2016*) are denoted with a **B-** followed by the compound numbers used in that work. (**B**) Bar graph showing activation of the ERSE.FLuc reporter in HEK293T cells treated with **147** (10 µM), thapsigargin (Tg; 0.5 µM) or the indicated dose of acetaminophen for 18 hr. Error bars show SEM for 3 independent experiments. (**C**) Immunoblot of PDI4 in **147**–**20** affinity purified proteins from HEK293T cells treated with **147** (10 µM), or **147**–**20** (10 µM), or the combination of **147**–**20** (10 µM) and **147** (50 µM), or the combination of **147**–**20** (10 µM) and **147**–**4** (50 µM) for 18 hr. The relative recovery of PDIA4 under these different conditions is indicated below the immunoblot.

DOI: https://doi.org/10.7554/eLife.37168.011

The following figure supplements are available for figure 4:

**Figure supplement 1.** Compound **147** selectively modifies ER-localized proteins.
DOI: https://doi.org/10.7554/eLife.37168.012

**Figure supplement 2.** Compound **147** preferentially partitions to the ER.
DOI: https://doi.org/10.7554/eLife.37168.013

both of which are largely cytosolic stress responses (*Trott et al., 2008*). Notably, celastrol does not activate the UPR ERSE.Fluc reporter in cells (*Cooley et al., 2014*). Furthermore, other compounds known to form reactive quinone methides such as eugenol, also did not significantly activate the ERSE.Fluc reporter (*Figure 4B* and *Figure 4—figure supplement 1*). Thus, the reaction preference

of **147** for protein targets in the ER (*Figure 4A*) and subsequent selective activation of the ATF6 UPR signaling pathway is a singular feature of the reactivity and specificity of this compound.

To further explore the origins of the selectivity of **147** and its analogs for modifying ER proteins, we sought to determine whether **147** preferentially partitions to the ER. We treated HEK293T cells with **147** for 15 min and separated membranous organelles including the ER from the cytosol by digitonin extraction (we validated that the digitonin extraction separated the ER and cytosol by blotting the fractions for the ER-localized ERdj3 and cytosolic HSC70; *Figure 4—figure supplement 2A*). We then used liquid chromatography-mass spectrometry (LC-MS) to quantify the amounts of **147** in the cytosol- and ER-containing fractions based on their peak areas by comparison with a standard curve. We found $2.1 \pm 0.2$ ng of **147** per million cells in the cytosol fraction compared to $5.6 \pm 0.9$ ng in the ER fraction (mean $\pm$ standard error; *Figure 4—figure supplement 2B*), corresponding to a ratio of $2.7 \pm 0.5$. Given that membranous organelles likely constitute a smaller fraction of the total volume of HEK293T cells than the cytosol (even in hepatocytes, membranous organelles are about only 40% of the total cell volume compared to 55% for the cytosol [*Weibel et al., 1969*]), this likely represents a lower limit for this ratio. This preference is likely due to the hydrophobicity of **147**, which would lead to its sequestration in the membranes that surround organelles like the ER.

This result suggests that the bias of **147** for reacting with ER-resident proteins is at least in part the result of its higher concentration in the ER and ER membrane. It is additionally worth noting that the cytochrome P450s that most likely convert **147** to **147-QM** (the quinone methide) primarily reside in the ER membrane (*Guengerich, 2015*). Moreover, the reducing environment of the cytosol would tend to decrease the lifetime of **147-QM** (e.g. through reaction with glutathione) relative to the lifetime of **147** in the oxidizing environment of the ER, thereby diminishing the extent of protein modification in the cytosol. The combination of these factors likely drives the preference of **147** to selectively react with ER-resident proteins.

Interestingly, despite preferential labeling of ER-localized PDIs by **147–20**, previous work has shown that treatment with **147** does not influence secretion of the endogenous secreted proteome or disulfide-bonded proteins dependent on PDIs for ER folding, such as energetically normal immunoglobulin light chains or fully-assembled immunoglobulins (*Plate et al., 2016*). This suggests that **147** may label only a fraction of the ER PDIs, allowing the remaining PDI population to be available for facilitating the folding of disulfide-bonded proteins within the ER lumen. To test this, we defined the fraction of PDIA4 labeled with **147–20** in HEK293T cells. We incubated HEK293T cells with **147–20** in the absence or presence of a 5-fold excess of the competitive ATF6 activator **147** or the inactive **147–4** (both lacking the means to attach an affinity handle) and then compared the recovery of PDIA4 in each purification to the total amount of PDIA4 in cell lysates. Only 24% of PDIA4 is covalently modified with **147–20** (*Figure 4C*). Conjugate formation by **147–20** is nearly eliminated when a 5-fold excess of the parent compound **147** is co-added, but not when the inactive compound **147–4** is co-added in excess. This suggests that while **147** prefers ER PDIs, it does not covalently modify the entire PDI population in the ER, providing an explanation for why treatment with **147** does not influence secretion of the endogenous secreted proteome or specific proteins dependent on PDIs for their efficient folding.

## shRNA-depletion of PDIs perturbs 147-dependent ATF6 activation

In the absence of stress, ATF6 is maintained in transport-incompetent monomeric and oligomeric conformations by intra- and intermolecular disulfide bonds (*Figure 5A*) (*Nadanaka et al., 2007*; *Nadanaka et al., 2006*; *Higa et al., 2014*). ER stress promotes the reduction of ATF6 disulfides, resulting in the formation of a reduced ATF6 monomer that traffics to the Golgi for proteolytic activation by the S1 and S2 proteases (*Nadanaka et al., 2007*; *Nadanaka et al., 2006*; *Higa et al., 2014*). Since the maintenance of disulfides in the ER requires PDI activity, the disulfide-mediated regulation of ATF6 also likely involves PDIs. Consistent with this hypothesis, previous results have shown that PDIA5 has a critical role in regulating ER stress-dependent ATF6 activation in cancer cells (*Higa et al., 2014*). These results suggest that **147**-dependent covalent modification of PDIs could promote ATF6 reduction and subsequent trafficking to the Golgi for activation (*Figure 5A*). Here, we probe this mechanism using a combination of pharmacologic and genetic approaches.

Initially, we tested this model by monitoring the reduction of FLAG-tagged ATF6 ([FT]ATF6) stably expressed in HeLa cells treated with **147** or the ER stressor tunicamycin (Tm). Oxidized and reduced [FT]ATF6 were separated by non-reducing and reducing SDS-PAGE/immunoblotting, as previously

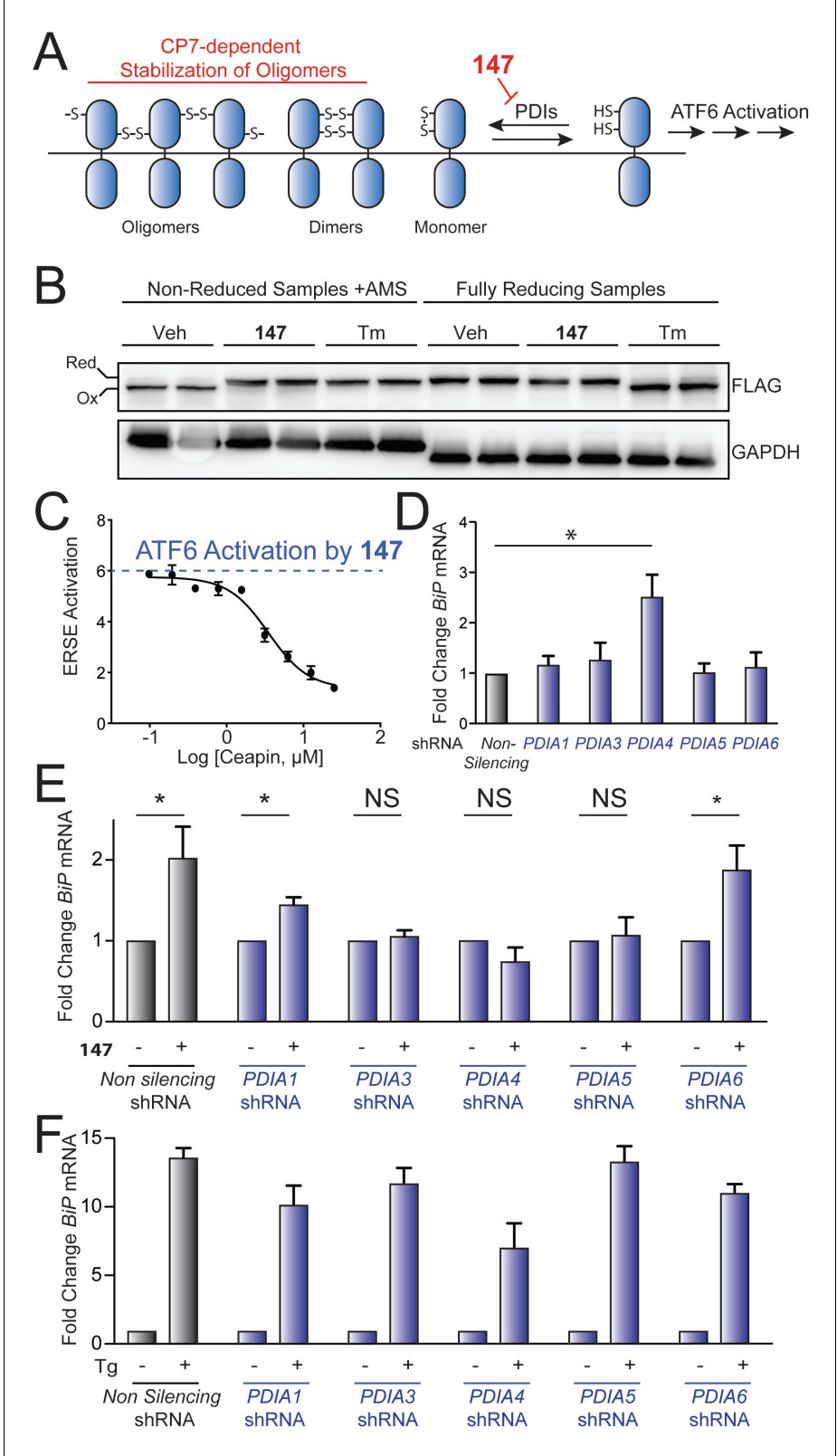

**Figure 5.** Genetic depletion of select PDIs alters **147**-dependent ATF6 activation. (**A**) Mechanistic model of **147**-dependent ATF6 activation, as described in the main text. The red component of the figure shows how pharmacologic modulation of ATF6 by **147** or CP7 influences ATF6 activity. (**B**) Immunoblot showing lysates prepared from HeLa cells stably expressing FLAG-tagged ATF6 treated for 6 hr with **147** (10 µM) or tunicamycin (Tm; 1 mg/mL). Lysates were separated by non-reducing or reducing SDS-PAGE prior to immunoblotting. The bands representing oxidized and reduced ATF6
*Figure 5 continued on next page*

*Figure 5 continued*

are indicated. Note the increased ATF6 migration observed in reducing gels for Tm-treated cells, reflecting the inhibited N-linked glycosylation of ATF6 in these samples. (C) Graph showing relative activation of the ERSE.Fluc ATF6 reporter in HEK293T cells co-treated for 18 hr with **147** (10 µM) and increasing concentrations of **CP7**, as indicated. Error bars show SEM for three technical replicates. (D) Bar graph showing *BiP* expression in HEK293T cells stably expressing non-silencing, *PDIA1*, *PDIA3*, *PDIA4*, *PDIA5*, or *PDIA6* shRNA, as indicated. *BiP* expression in the PDI-depleted cells is shown relative to cells expressing non-silencing shRNA. Error bars show SEM for 3 independent experiments. *indicates p<0.05. (E) Bar graph showing *BiP* expression in HEK293T cells expressing non-silencing, *PDIA1*, *PDIA3*, *PDIA4*, *PDIA5*, or *PDIA6* shRNA treated for 6 hr with or without **147** (10 µM). *BiP* expression levels for samples treated with **147** are normalized to the corresponding vehicle-treated controls. Un-normalized data are shown in *Figure 5—figure supplement 1B*. Error bars show SEM for 3 independent experiments. *indicates p<0.05. (F) Bar graph showing *BiP* expression in HEK293T cells expressing non-silencing, *PDIA1*, *PDIA3*, *PDIA4*, *PDIA5*, or *PDIA6* shRNA treated for 6 hr with or without thapsigargin (Tg; 0.5 µM). *BiP* expression levels for samples treated with Tg are normalized to the corresponding vehicle-treated controls. Un-normalized data are shown in *Figure 5—figure supplement 1C* Error bars show SEM for 3 independent experiments.

DOI: https://doi.org/10.7554/eLife.37168.014

The following figure supplements are available for figure 5:

**Figure supplement 1.** Genetic depletion of select PDIs disrupts **147**-dependent ATF6 activation.

DOI: https://doi.org/10.7554/eLife.37168.015

**Figure supplement 2.** Structure of **132**.

DOI: https://doi.org/10.7554/eLife.37168.016

---

described (*Nadanaka et al., 2007*). In the absence of small molecule treatment, [FT]ATF6 migrates faster in non-reducing gels, consistent with its primarily oxidized status (*Figure 5B*). Treatment with **147** alters the migration of [FT]ATF6 in non-reducing gels (upward gel shift), consistent with a net reduction of the disulfides in ATF6. Similar results were observed upon Tm treatment, which activates all 3 arms of the UPR. These experiments support the hypothesis that **147**-dependent covalent conjugation to PDIs promotes the reduction of ATF6 to increase populations of the reduced ATF6 monomers associated with transcription factor activation (*Plate et al., 2016*).

A set of small molecules called Ceapins were previously identified to inhibit ER stress-dependent ATF6 activation by stabilizing ATF6 oligomers in the ER and preventing their trafficking to the Golgi for proteolytic activation (*Gallagher and Walter, 2016*; *Gallagher et al., 2016*). Based on the proposed model shown in *Figure 5A*, we predict that stabilization of ATF6 oligomers by Ceapins should impede **147**-dependent ATF6 activation by preventing the disruption of the ATF6 oligomers into the reduced monomers required for trafficking. Consistent with this, **147**-dependent activation of the ATF6-selective ERSE.Fluc reporter was inhibited by co-addition of increasing concentrations of the active Ceapin, **CP7** (*Figure 5C*; blue dotted line indicates level of **147** activation achieved in the absence of **CP7**) (*Gallagher and Walter, 2016*; *Gallagher et al., 2016*). This indicates that stabilization of ATF6 oligomers disrupts **147**-dependent ATF6 activation, consistent with **147** having an important role in promoting ATF6 disulfide reduction and ATF6 dissociation prior to trafficking to the Golgi for proteolytic activation.

Next, we used a genetic approach to define the dependence of **147**-mediated ATF6 activation on specific PDIs identified in our quantitative mass spectrometry experiment. We prepared stable pools of HEK293T cells wherein *PDIA1*, *PDIA3*, *PDIA4*, or *PDIA6* was shRNA-depleted to extents analogous to the population of PDIAs covalently labeled by **147** (see *Figure 4C*). We also prepared a stable pool of HEK293T cells shRNA-depleted of *PDIA5*, which was identified as modified by **147–20** in 2 out of 3 cell types (*Table 1—source data 2*) and has previously been shown to be important for ATF6 activation (*Higa et al., 2014*). PDI knockdown in these cells varied from 25–50%, as measured by qPCR (*Figure 5—figure supplement 1A*), consistent with the population of each PDI predicted to be labeled by **147** (see *Figure 4C*). Knockdown of *PDIA4* increased expression of the ATF6 target gene *BiP* in untreated cells, indicating that reduced PDIA4 activity activates ATF6 independently of compound addition (*Figure 5D*). However, knockdown of other PDIs did not significantly influence *BiP* expression. The addition of **147** did not increase expression of the ATF6-regulated gene *BiP*, relative to matched untreated controls, in cells in which *PDIA3*, *PDIA4*, or *PDIA5* were depleted by shRNA (*Figure 5E*; unnormalized data are shown in *Figure 5—figure supplement 1B*). In contrast, Tg-dependent induction of *BiP* was not significantly impaired in these cells upon PDI knockdown (*Figure 5F*; unnormalized data are shown in *Figure 5—figure supplement 1C*). Interestingly, shRNA-depletion of *PDIA6* did not perturb *BiP* expression in **147**- or Tg-treated cells,

suggesting that covalent modification of this PDI is not required for ATF6 activation in these cells. Similarly, *PDIA1* depletion only modestly reduced **147**-dependent *BiP* expression, indicating that covalent modification of PDIA1 also does not significantly influence ATF6 activation in these cells. Collectively, these results show that modest depletion (25–50%) of a subset of PDIs selectively inhibits **147**-dependent activation of ATF6, suggesting that covalent modification of PDIA3, PDIA4, and/or PDIA5 contribute to **147**-dependent ATF6 activation.

The notion that **147** acts through modification of PDIs, some of which are in turn upregulated by **147**-mediated ATF6 activation (especially PDIA4, and to a lesser extent PDIA1 and PDIA3) (*Plate et al., 2016*) raises the question of whether the increased levels of PDIs could diminish the efficacy of **147** as time goes on. ATF6 activity induced by **147** was previously shown to diminish 12 hr after a single treatment of compound in HEK293 cells, suggesting a potential negative feedback loop associated with ATF6 activation (*Plate et al., 2016*). However, **147** has been used in experiments where it was shown that daily dosing of **147** over weeks could suppress stem cell pluripotency, enhance differentiation, and direct mesodermal cell fate (*Kroeger et al., 2018*), suggesting that the induction of PDIs does not lead to a substantial feedback attenuation of **147**-dependent ATF6 activation when constantly administered. Thus, while the potential for increased PDI levels to suppress **147**-dependent ATF6 activation offers an interesting mechanism to modulate **147** activity, further studies will be required to elucidate whether such a negative feedback mechanism is operating.

## Discussion

Here, we hypothesize that ATF6 activation by **147** proceeds through a mechanism involving metabolic oxidation of **147** to form a reactive compound, likely a *p*-quinone methide, that covalently modifies a subset of ER-localized proteins, including PDIs. This targeting of PDIs by **147** appears to disrupt the maintenance of ATF6 disulfides, enabling reduction of disulfides within ATF6, increasing the populations of reduced, trafficking-competent ATF6 monomers. Previous results show that ATF6 reduction is not sufficient to promote the complete trafficking of ATF6 to the Golgi (*Nadanaka et al., 2007*; *Nadanaka et al., 2006*). However, increasing populations of reduced, monomeric ATF6 would enhance sensitivity of this transcriptional signaling pathway to normal, homeostatic changes in ER proteostasis, thus increasing basal activity of ATF6 transcriptional signaling (*Figure 5A*). This mechanism would explain the 30–40% activation of ATF6 afforded by **147** (*Plate et al., 2016*) and highlights the potential for targeting ATF6 redox to allow for the moderate levels of ATF6 signaling necessary to therapeutically modulate ER proteostasis without impacting global ER function (*Plate and Wiseman, 2017*).

A remarkable aspect of this mechanism of pharmacologic ATF6 activation is the preferential covalent modification of proteins localized to the ER by **147**. This preference likely results from the biased partitioning of **147** to the ER and other membranous compartments of the cell due to its hydrophobic nature as well as the oxidation of **147** by cytochrome P450s localized to the ER membrane. The conversion of **147** to reactive compounds such as a *p*-quinone methide at the ER membrane limits the 'sphere of activity' for this compound, allowing it to selectively modify ER proteins, like PDIs acting on ATF6, similarly localized to the ER membrane. In this scenario, the reactive **147-QM** cannot diffuse far from the ER before it hydrolyzes or reacts with small molecule cellular nucleophiles (e.g. glutathione) that prevent the broad modification of cellular proteins localized to other compartments. This is especially true in the cytosol, in which **147-QM** likely has a shorter lifetime than in the ER because of the cytosol's more reducing environment.

This mechanism of activity for **147** is enabled by a coincidence of several structural features of the small molecule. First, the 2-amino-*p*-cresol substructure of **147** can be oxidized to a *p*-quinone methide, a highly efficient electrophile for covalent modification of proteins. Second, each of the three primary substructures of **147**, the A-ring, B-ring, and three carbon linker (*Figure 1A*), are largely required for compound activity. The 2-amino-*p*-cresol substructure of the A-ring is required for **147** oxidation, for example quinone methide formation. However, the dependence on the B-ring and linker for ATF6 activation remains unclear. These substructures could be important for targeting the compound to a specific subset of cytochrome P450s at the ER membrane for metabolic activation and/or directing the reactive, oxidized **147-QM** to PDIs for covalent modification, or both. While we continue to define the dependence of **147** activity on these substructures, our results

show that the B-ring and linker confer the selectivity of **147** required for compound-dependent ATF6 activation. The third feature of **147** important for ATF6 activation is the unique reactivity of the resulting oxidized product. The reactivity of the metabolically activated **147** is not so high that it is instantly quenched by hydrolysis, nor so low that it can diffuse away from the immediate ER environment and react widely with proteins localized throughout the cell.

Collectively, these features combine to make **147** a precision electrophile that preferentially activates the ATF6 signaling pathway without triggering other pathways activated by similar stimuli, such as the IRE1 and PERK UPR signaling pathways, the heat shock response, or the NRF2 oxidative stress response as we have shown previously by multiplex gene expression profiling, as well as by RNAseq (*Plate et al., 2016*). ER localization enabling localized generation of electrophiles by P450s and related enzymes that selectively modify protein(s) is a potentially interesting drug development strategy. However, it is important to note that in our previous work we also found that some compounds with similar structures, notably including the 2-amino-*p*-cresol warhead, are substantially less selective towards activating stress-responsive signaling. The small molecule **132**, shown in *Figure 5— figure supplement 2*, activates ATF6, XBP1, and PERK stress sensors to similar extents (*Plate et al., 2016*). Given their structural similarity, it is reasonable to expect that **132** is oxidized to a quinone methide followed by proteome reactivity like **147**. Thus, it appears that pushing reactivity too far and/or quinone methide concentration too high can result in a broader stress-responsive signaling pathway activation beyond ATF6. The PDIs that **147** modifies could interact with other components of the ER proteostasis network, in addition to ATF6 (*Gallagher et al., 2016*; *Ryno et al., 2014*). The selectivity of **147** likely arises from it having a lower propensity to modify the Cys-containing proteome than **132**, either because **147** is not as reactive toward protein nucleophiles, or because its quinone methide is at lower concentration, or because it is more rapidly quenched.

Pharmacologic ATF6 activation has substantial potential for ameliorating tissue-specific defects in ER proteostasis implicated in diverse diseases including systemic amyloid diseases, ischemic heart disease, diabetes, and neurodegenerative disorders (*Hetz et al., 2015*; *Plate and Wiseman, 2017*). The mechanism of action for our top pharmacologic activator **147** (as defined herein) suggests new potential opportunities to optimize the activity of pharmacologic ATF6 activators for specific tissues. For example, designing **147** analogs with selectivity for a subset of P450s provides a potential opportunity to improve compound activity in tissues specifically expressing this subset of P450s. Similarly, different tissues have different PDI compositions. Thus, improving targeting of reactive compounds for specific PDIs could improve ATF6 activating activity in tissues expressing these PDIs. Interestingly, this type of selectivity for P450s or PDIs provides a potential explanation for how other compounds identified in our original HTS that can be similarly metabolically activated to reactive compounds (e.g. **263**) efficiently activate ATF6 in some cell lines (e.g. HEK293T), but not in others (e.g. light chain amyloidosis patient-derived ALMC2 cells) (*Plate et al., 2016*). Thus, the mechanism defined herein suggests new opportunities to selectively promote ATF6 activation in tissues of interest by further development of these compounds through medicinal chemistry efforts.

## Materials and methods

### Chemicals

All compounds and reagents were purchased from Sigma-Aldrich, Acros, Alfa Aesar, and EMD Millipore unless otherwise noted and were used without further purification. The ERSE-Firefly luciferase reporter cell line has been reported previously (*Plate et al., 2016*) and was used without modification.

### Cell culture and transfections

HEK293T-Rex (ATCC), HEK293T (ATCC), and HepG2 (ATCC) were cultured in high-glucose Dulbecco's Modified Eagle's Medium (DMEM) supplemented with glutamine, penicillin/streptomycin and 10% fetal bovine serum (FBS). Cells were routinely tested for mycoplasma every 6 months. We did not further authenticate the cell lines. HEK293T-Rex cells containing the ERSE-FLuc reporters were created by transfection with ERSE-FLuc pcDNA3.1 by calcium phosphate followed by culturing in geneticin sulfate (G-418, 500 µg/mL). All cells were cultured under typical tissue culture conditions (37°C, 5% $CO_2$).

## Measurement of ATF6 activity using the ERSE.Fluc reporter

HEK293T-Rex cells incorporating the ERSE.FLuc reporter were plated at 100 µL/well from suspensions of 200,000 cells/mL in black clear-bottom 96-well plates (Corning) and incubated at 37°C overnight. The following day, cells were treated with 10 µM of compound for 18 hr at 37°C or were pretreated with the necessary compound as described before addition of compounds for 18 hr at 37°C. The plates were equilibrated to room temperature, then 100 µL of Firefly luciferase assay reagent-1 (Targeting Systems) were added to each well. Luminescence activity was measured 10 min after reagent addition with an EnVision Multilabel Reader (PerkinElmer) using a 100 ms integration time.

## SDS-PAGE-based analysis of proteins modified by 147

Affinity-precipitated samples (see below) for SDS-PAGE were normalized for protein concentration using the BCA assay (Thermo Fisher). Protein samples were boiled for 5 min in Laemmli buffer + 100 mM DTT before loading onto a 10% SDS-PAGE gel. The gel was fixed in 40% methanol/10% acetic acid solution. The gel was then stained with 0.1% Coomassie Blue R250 in 10% acetic acid/50% methanol solution. Prominent bands were excised and the proteins were extracted, digested and analyzed by mass spectrometry.

## Immunoblotting

For immunoblotting, cells were lysed in 50 mM Tris buffer, pH 7.5 containing 0.1% TritonX (Fisher Scientific) and supplemented with protease inhibitor cocktail (Roche). Protein lysate concentrations were normalized by BCA assay (Thermo Fisher). Lysates were boiled for 5 min in Laemmli buffer with 100 mM DTT before loading onto SDS-PAGE gels. Proteins were transferred from gel slabs to nitrocellulose and blotted using mouse anti-FLAG M2 antibody (Sigma) or rabbit anti-PDIA4 antibody (Protein Tech) and visualized on the Odyssey Infrared Imaging System (Li-Cor Biosciences).

## Affinity precipitation of proteins covalently modified by 147

HEK293T-Rex cells in 10 cm plates were treated for 16 hr with **147** (10 µM), or **147–20** (10 µM), or the combination of **147–20** (10 µM) and **147** (50 µM), or **147–20** (10 µM) in combination with **147–4** (50 µM) at 37°C. Lysates were prepared in radioimmunoprecipitation assay (RIPA) buffer (150 mM NaCl, 50 mM Tris pH 7.5, 1% Triton X-100, 0.5% sodium deoxycholate, and 0.1% SDS) with fresh protease inhibitor cocktail (Roche, Indianapolis, IN) and centrifuged for 20 min at 10,000 × *g*. Protein concentrations of supernatants were determined by the BCA assay (Thermo Fisher). For each sample, 100 µg of lysate were reacted with click reagents to give final concentrations as follows: 100 µM of diazo biotin-azide (Click Chemistry Tools, Scottsdale, AZ), 800 µM copper (II) sulfate, 1.6 mM BTTAA ligand (2-(4-((bis((1-tert-butyl-1H-1,2,3-triazol-4-yl)methyl)amino)methyl)−1 H-1,2,3-triazol-1-yl)acetic acid) (Albert Einstein College), and 5 mM sodium ascorbate. The reaction was placed on a shaker at 1000 rpm at 30°C for 2 hr. The proteins were then precipitated from the reaction mixture by adding an equal volume of 3:1 chloroform/methanol. The pellet was washed three times with 1:1 chloroform/methanol. The precipitate was suspended in 500 µL of 6 M urea with 25 mM ammonium bicarbonate and 140 µL of 10% SDS was added to this mixture to help solubilize the protein. 50 µL of high capacity streptavidin beads were washed with PBS and mixed with the protein solution in 6 mL of phosphate-buffered saline (PBS). This suspension was placed on a rotator or a shaker and agitated for 2 hr. The beads were centrifuged and washed five times with PBS and 1% SDS. The protein was eluted from the beads by two washes of 50 mM sodium dithionite in 1% SDS for 1 hr and then precipitated by chloroform/methanol precipitation as described above.

## TMT-MuDPIT proteomics-based identification of proteins modified by 147

Air-dried pellets from the affinity precipitation were resuspended in 1% RapiGest SF (Waters) in 100 mM HEPES (pH 8.0). Proteins were reduced with 5 mM tris(2-carboxyethyl)phosphine hydrochloride (Thermo Fisher) for 30 min and alkylated with 10 mM iodoacetamide (Sigma Aldrich, St. Louis, MO) for 30 min at ambient temperature and protected from light. Proteins were digested for 18 hr at 37°C with 2 µg trypsin (Promega). After digestion, 20 µg of peptides from each sample were reacted for 1 hr with the appropriate TMT-NHS isobaric reagent (ThermoFisher) in 40% (v/v) anhydrous acetonitrile and quenched with 0.4% NH$_4$HCO$_3$ for 1 hr. Samples with different TMT labels were pooled

and acidified with 5% formic acid. Acetonitrile was evaporated on a SpeedVac and debris was removed by centrifugation for 30 min at 18,000 × g. MuDPIT (Multi-Dimensional Protein Identification Technology) microcolumns were prepared as described previously (*Ryno et al., 2014*). LCMS/MS analysis was performed using a Q Exactive mass spectrometer equipped with an EASY nLC 1000 (Thermo Fisher). MuDPIT experiments were performed by 5 min sequential injections of 0, 20, 50, 80, 100% buffer C (500 mM ammonium acetate in buffer A) and a final step of 90% buffer C/10% buffer B (20% water, 80% acetonitrile, 0.1% fomic acid, v/v/v) and each step followed by a gradient from buffer A (95% water, 5% acetonitrile, 0.1% formic acid) to buffer B. Electrospray ionization was performed directly from the analytical column by applying a voltage of 2.5 kV with an inlet capillary temperature of 275°C. Data-dependent acquisition of MS/MS spectra was performed with the following settings: eluted peptides were scanned from 400 to 1800 m/z with a resolution of 30,000 and the mass spectrometer in a data dependent acquisition mode. The top ten peaks for each full scan were fragmented by HCD using a normalized collision energy of 30%, a 100 ms activation time, a resolution of 7500, and scanned from 100 to 1800 m/z. Dynamic exclusion parameters were 1 repeat count, 30 ms repeat duration, 500 exclusion list size, 120 s exclusion duration, and exclusion width between 0.51 and 1.51. Peptide identification and protein quantification was performed using the Integrated Proteomics Pipeline Suite (IP2, Integrated Proteomics Applications, Inc., San Diego, CA) as described previously (*Ryno et al., 2014*).

## Measurement of 147 concentration in ER versus cytosol

The concentration of **147** in the ER was determined by treating HEK293T cells for 15 min with 10 µM **147**. The cells were scraped off the plate with 1 mM EDTA in TBS and the cellular pellet was lysed with 0.1% digitonin in HEPES (pH 7.5), 100 mM NaCl. The supernatant (cytosol) was analyzed for enrichment of cytosolic proteins (HSC70) and the pellet (ER) was analyzed for ER proteins (ERdj3) by immunoblot. To determine the concentrations of **147** in the ER and cytosol, each enriched fraction was subject to a chloroform: methanol extraction by adding 4× volume of both chloroform and methanol to precipitate the proteins. The organic phase was collected, concentrated, and resuspended in methanol. **147** concentrations were determined using a Waters TQ-XS triple quad mass spectrometer by injecting 5 µL of sample onto a Waters HSS T3 C18 column (2.1 × 10 mm, 1.8 µM) with a flow rate of 0.3 mL per minute and comparing the observed peak areas to a calibration curve.

## Measurement of ATF6 oxidation state

The oxidation state of ATF6 in HeLa cells treated with **147** was analyzed by gel-shift essentially as previously described (*Nadanaka et al., 2006*). Briefly, cells were infected with adenovirus encoding FLAG-tagged full length inactive ATF6 [ATF6(1–670)] and subsequently treated with respective treatments for 6 hr. Cells were lysed in low-stringency lysis buffer comprising 20 mM Tris-HCl (pH 7.5), 150 mM NaCl, 1% Triton X-100, protease inhibitor cocktail (Roche Diagnostics, Indianapolis, IN) and phosphatase inhibitor cocktail (Roche Diagnostics) and 20 µM 4-Acetamido-4′-Maleimidylstilbene-2,2′-Disulfonic Acid, Disodium Salt (AMS) (Thermo Fisher, cat# A485). AMS binds covalently to reduced thiols, typically on cysteine residues, and increases their molecular mass in SDS-PAGE. Thus, proteins that exhibit an upward shift when analyzed under non-reducing conditions compared to reducing are considered to have reduced thiols.

## shRNA knockdown of PDIs and subsequent measurement of ATF6 activation by 147

HEK293T-Rex (Invitrogen) cells were cultured in complete DMEM (CellGro) supplemented with 10% FBS (CellGro) and penicillin/streptomycin (CellGro). Lentiviruses encoding a mix of shRNAs that are either non-silencing or directed against *PDIA1*, *PDIA3*, *PDIA4*, *PDIA5*, or *PDIA6* were transduced into HEK293T-Rex with 1–5 mL of virus in media containing 5 mg/mL polybrene (see *Supplementary File 1* for lentivirus production methods). Stable cell lines were selected by culturing in puromycin (5 µg/mL) before characterization. The expression of the ATF6 target gene *BiP* in cells treated with **147** (10 µM) or thapsigargin (0.5 µM) was then measured by qPCR as described previously.

## Quantitative RT-PCR

The relative mRNA expression levels of target genes were measured using quantitative RT-PCR (see *Supplementary File 1* for RT-PCR details and sequences of primers used).

## Statistical methods

The data in *Figure 2* and *Figure 5* were tested for significance using ANOVA with a post-hoc Dunnett's test. The PANTHER overrepresentation test has been described elsewhere (*Mi et al., 2017*) and was used as implemented by the PANTHER website (http://pantherdb.org/) using Fisher's exact test with multiple testing correction.

## Synthesis and characterization

Details of the synthesis and characterization of **147** and its derivatives are reported in *Supplementary File 1*.

# Additional information

### Competing interests

Ryan Paxman, Lars Plate, Evan T Powers, R Luke Wiseman: has submitted a patent application (WO2017117430A1) for the use of 147 and other compounds as ER proteostasis network regulators to treat protein misfolding diseases. Jeffery W Kelly: is a co-founder and member of the Scientific Advisory Board of Proteostasis Therapeutics Inc., who is independently pursing ATF6 activators; however, he is unaware of the structures of their activators, which were discovered by a totally different screening approach. Has submitted a patent application (WO2017117430A1) for the use of 147 and other compounds as ER proteostasis network regulators to treat protein misfolding diseases. The other authors declare that no competing interests exist.

### Funding

| Funder | Grant reference number | Author |
|---|---|---|
| National Institutes of Health | AG046495 | R Luke Wiseman Jeffery W Kelly |
| National Institutes of Health | DK107604 | R Luke Wiseman |
| National Institutes of Health | HL075573 | Chris Glembotski |
| National Institutes of Health | HL104535 | Chris Glembotski |
| National Institutes of Health | HL085577 | Chris Glembotski |
| Leukemia and Lymphoma Society | 5439-16 | Lars Plate |
| American Heart Association | 17PRES33670796 | Erik A Blackwood |
| San Diego State University | Rees-Stealy Research Foundation | Erik A Blackwood |
| San Diego State University | Heart Institute Inamori Foundation | Erik A Blackwood |
| Achievement Rewards for College Scientists Foundation | San Diego chapter | Erik A Blackwood |

The funders had no role in study design, data collection and interpretation, or the decision to submit the work for publication.

### Author contributions

Ryan Paxman, Conceptualization, Investigation, Formal analysis, Data curation, Writing-original draft, Writing—review and editing; Lars Plate, Conceptualization, Investigation, Formal analysis, Data curation, Writing—original draft, Writing—review and editing; Erik A Blackwood, Investigation, Data curation, Formal analysis, Writing—review and editing; Chris Glembotski, Project Administration,

Funding acquisiton, Writing—review and editing; Evan T Powers, Conceptualization, Formal analysis, Data curation, Writing-original draft, Writing—review and editing; R Luke Wiseman, Conceptualization, Formal analysis, Data curation, Writing-original draft, Writing—review and editing, Supervision, Project Administration, Funding acquisition; Jeffery W Kelly, Conceptualization, Project administration, Supervision, Funding acquisition, Writing-original draft, Writing—review and editing

## Author ORCIDs
Ryan Paxman (iD) https://orcid.org/0000-0001-6421-1892
Lars Plate (iD) https://orcid.org/0000-0003-4363-6116
Evan T Powers (iD) https://orcid.org/0000-0001-8185-8487
R Luke Wiseman (iD) https://orcid.org/0000-0001-9287-6840
Jeffery W Kelly (iD) https://orcid.org/0000-0001-8943-3395

## Decision letter and Author response
Decision letter https://doi.org/10.7554/eLife.37168.021
Author response https://doi.org/10.7554/eLife.37168.022

## Additional files

### Supplementary files
• Supplementary file 1. Supplementary experimental methods, including lentivirus production, qPCR primers and synthetic methods and characterization data for small molecules.
DOI: https://doi.org/10.7554/eLife.37168.017

• Transparent reporting form
DOI: https://doi.org/10.7554/eLife.37168.018

### Data availability
All data generated or analyzed during this study are included in the manuscript and supporting files.

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
