## [Decision Letter]

Thank you for submitting your article "A Small Molecule that Preferentially Activates the ATF6 Pathway of the Unfolded Protein Response is Metabolically Activated to Selectively Modify Endoplasmic Reticulum Proteins" for consideration by *eLife*. Your article has been reviewed by two peer reviewers, and the evaluation has been overseen by a Reviewing Editor and Randy Schekman as the Senior Editor. The reviewers have opted to remain anonymous.

The reviewers have discussed the reviews with one another and the Reviewing Editor has drafted this decision to help you prepare a revised submission.

Summary:

The authors present an excellent follow up study to their previous *eLife* report (Plate et al., 2016) of a novel small molecule compound that specifically upregulates the ATF6 pathway, one of the three arms of the unfolded protein response. Because pharmacological modulation of stress pathways is a promising avenue of intervention for numerous diseases, the molecule called **147**, was proposed to have potential therapeutic value. However, the mechanism of activation was unclear in the previous study. In the current study, the authors took an exhaustive approach to understand how **147** activates the pathway. The mechanism involves the modification of **147** via enzymatic oxidation by cytochrome P450. Quite surprisingly, activated **147** then preferentially reacts with ER proteins, particularly protein disulfide isomerases (PDIs). Because PDIs are targets of the UPR and can play a role in stress regulation, the finding was intriguing but potentially troubling if **147** or a similar derivative is to be used as a therapeutic. However, previous work suggests that **147** does not cause detrimental effects to the assembly and secretion of molecules examined.

In summary, it is the consensus view of reviewers that this manuscript represents an important advancement to the previous study. The experimental designs are well conceived and experiments well performed and presented. The text is well written and remarkably approachable for a topic that might otherwise be challenging to the broad readership of *eLife*. The reviewer recommendation is accept with minor revisions.

Essential revisions:

1) One of the big remaining mysteries is why **147** labels primarily ER proteins. This is interesting because other efforts have generated molecules that localize to the mitochondria (from P. Wipf, S. Kelley, etc.), but there is less known about ER localization. A few quick experiments might be warranted here, using existing tools. First, does **147-20** actually localize to the ER or is it just preferentially activated there? The experiment is a centrifugation separation of the ER compartment and a comparison of the levels of **147-20** in the ER and cytoplasm to measure relative enrichment. If this experiment was performed early after treatment, little covalent modification will have taken place and the ratio of enrichment might be determined. The other experiment is to understand if the P450-mediated activation to the quinone takes place under reducing conditions in vitro? One simple model is that the **147**reactivity is quenched by other reducing agents (glutathione) in other compartments (cytoplasm, nucleus), but the unique oxidizing environment of the ER (Birk et al., 2013, J. Cell Sci. 126:1604) allows reactivity.

2) The other interesting question is whether **147** treatment elevates PDIA levels through ATF6 activation over time? In other words, does the ATF6 activation lead to a shut-off of the compound activity by increasing the receptor level? In the author's previous work, I think they found that ATF6 activates PDIA4 and PDIA6 (Shoulders et al., 2013). This part of the mechanism is potentially important for the use of **147** in longer term treatment regimes and it might explain the relatively low toxicity.

3) The evidence for apparent selectivity of ER proteins and specifically 6 PDIs by **147** is convincing, and further supported by the genetic evidence using shRNA knockdown that PDIA3, PDIA4, and PDIA5 affect the **147**-dependent regulation of ATF6. The basis for specificity of these oxidized PDIs and perhaps other identified targets on regulation of ATF6 is not examined or much discussed. Presumably the effect of **147** is not due to conformational effects, i.e. causing protein damage and generation of misfolded proteins in the ER, however some of these PDIs are known to interact with calreticulin and other components of the lumen chaperone machinery. This needs to be addressed, either experimentally or minimally by a more thorough discussion as the stress signals for XBP1 and HSF1 are directly linked to increased flux of misfolded proteins. Alternatively, these data might suggest a regulatory process involving electrophile stress similar to Nrf2/Keap1.

*Reviewer #1:*

This work is an extension of studies to identify arm-selective activators of the UPR. In the prior work, a series of high throughout screens and secondary assays was used to identify compounds, including **147**, that activate the ATF6 pathway. Here, the authors take the next step – identifying the target and mechanism-of-action (MoA) of compound **147**. This is an important step because any phenotypic screen will provide active molecules, but knowledge of the target(s) and MoA is critical to utility and advancement as a chemical probe.

In this work, the authors start with an analog effort that is designed to reveal the pharmacophore. Through ~20 molecules, they quickly find that the 2-amino-*p*-cresol is absolutely required for activity in an ATF6 reporter cell line. The NMR and full chemical characterization is included in supporting information, and the purity and identity look appropriate. One of the key findings of the SAR study was the requirement for the methyl group opposite the 2-amino phenol, which strongly suggests a quinone tautamerization (as in eugenol and acetaminophen, Figure 2—figure supplement.1). Indeed, many pieces of evidence supported this idea, including co-treatment with a P450 inhibitor or BME (Figure 2). The authors take advantage of this mechanism and the prior SAR to create an alkyne-containing Click probe (**147-20**) that could be used to covalently modify the relevant cellular targets (Figure 3). The authors use all the right controls in this tricky proteomics experiment, including using 5-fold excess of **147** as a competitor and **147-4** as a negative control. This effort pointed to PDIs as likely targets, among a handful of other proteins (Table 1). The enrichment of ER proteins was a big surprise to this reviewer, as there did not seem to be any structural feature of **147** that might cause it to localize in the ER (see below). Next, the authors perform important studies using knockdown of PDIs to show that PDIA3, PDIA4 and PDIA5 seems to be especially important in ATF6 activation (Figure 5D) and for **147** activity (Figure 5E), suggesting how **147** works through these proteins. Again, these studies are comprehensive, with transcriptional support for the mechanism.

This is an interesting study that represents a mountain of difficult work. It advances the first well-characterized inhibitors of PDIs and some of the first ATF6 activators. Finally, it is the logical next step from the previous *eLife* manuscript. I support publication with a handful of clarifications (below).

1) One of the big remaining mysteries is why **147** labels primarily ER proteins. This is interesting because other efforts have generated molecules that localize to the mitochondria (from P. Wipf, S. Kelley, etc.), but there is less known about ER localization. A few quick experiments might be warranted here, using existing tools. First, does **147-20** actually localize to the ER or is it just preferentially activated there? The experiment is a centrifugation separation of the ER compartment and a comparison of the levels of **147-20** in the ER and cytoplasm to measure relative enrichment. If this experiment was performed early after treatment, little covalent modification will have taken place and the ratio of enrichment might be determined. The other experiment is to understand if the P450-mediated activation to the quinone takes place under reducing conditions in vitro? One simple model is that the **147**reactivity is quenched by other reducing agents (glutathione) in other compartments (cytoplasm, nucleus), but the unique oxidizing environment of the ER (Birk et al., 2013, J. Cell Sci. 126:1604) allows reactivity.

2) The other interesting question is whether **147** treatment elevates PDIA levels through ATF6 activation over time? In other words, does the ATF6 activation lead to a shut-off of the compound activity by increasing the receptor level? In the author's previous work, I think they found that ATF6 activates PDIA4 and PDIA6 (Shoulders et al., 2013). This part of the mechanism is potentially important for the use of **147** in longer term treatment regimes and it might explain the relatively low toxicity.

*Reviewer #2:*

This paper examines the mechanism by which N-(2-hydroxy-5-methylphenyl)-3-phenylpropanamide (hereafter **147**), previously identified by the Kelly lab, activates the ATF6 arm of the UPR. The authors show that **147** is oxidized by cytochrome P450s to generate an ER localized electrophile that reacts preferentially with a set of mostly ER-localized proteins including protein disulfide isomerases that are proposed to regulate ATF6 activity.

This is a clearly presented paper and the chemistry for specificity of **147** on activation of ATF6 appears thorough and well executed. The underlying chemistry of **147** and its use as a chemical probe is intriguing. The data show that the effects of **147** are essentially ER compartment specific; that other cell stress responses such as the HSR are not induced and from the proteomics data that mostly ER localized proteins are targets of **147**. While the logic makes sense, there is a gap in the story to understand how oxidation of specific PDIs by **147** leads to activation ATF6. Nevertheless, there are many aspects of this paper that are worthy of publication.

The evidence for apparent selectivity of ER proteins and specifically 6 PDIs by **147** is convincing, and further supported by the genetic evidence using shRNA knockdown that PDIA3, PDIA4, and PDIA5 affect the **147**-dependent regulation of ATF6. The basis for specificity of these oxidized PDIs and perhaps other identified targets on regulation of ATF6 is not examined or much discussed. Presumably the effect of **147** is not due to conformational effects, i.e. causing protein damage and generation of misfolded proteins in the ER, however some of these PDIs are known to interact with calreticulin and other components of the lumen chaperone machinery. This needs to be addressed, either experimentally or minimally by a more thorough discussion as the stress signals for XBP1 and HSF1 are directly linked to increased flux of misfolded proteins. Alternatively, these data might suggest a regulatory process involving electrophile stress similar to Nrf2/Keap1.

---

## [Author Response]

Reviewer #1:[…] 1) One of the big remaining mysteries is why **147** labels primarily ER proteins. This is interesting because other efforts have generated molecules that localize to the mitochondria (from P. Wipf, S. Kelley, etc.), but there is less known about ER localization. A few quick experiments might be warranted here, using existing tools. First, does **147-20** actually localize to the ER or is it just preferentially activated there? The experiment is a centrifugation separation of the ER compartment and a comparison of the levels of **147-20** in the ER and cytoplasm to measure relative enrichment. If this experiment was performed early after treatment, little covalent modification will have taken place and the ratio of enrichment might be determined. The other experiment is to understand if the P450-mediated activation to the quinone takes place under reducing conditions in vitro? One simple model is that the **147**reactivity is quenched by other reducing agents (glutathione) in other compartments (cytoplasm, nucleus), but the unique oxidizing environment of the ER (Birk et al., 2013, J. Cell Sci. 126:1604) allows reactivity.

We agree with the reviewer that the origin of the selectivity of **147** for ER proteins is an important unanswered question. To determine whether **147** and its analogs preferentially react with ER proteins because they localize to the ER, we performed the insightful experiment that the reviewer suggests. We treated HEK293T cells with **147** for 15 minutes and separated the ER and other membranous organelles from the cytosol by using a digitonin extraction (we validated that the digitonin extraction for the most part separated the ER and cytosol by blotting the fractions for ERdj3 and HSC70. As expected, ERdj3 and Hsc70 were enriched in the membranous and cytosolic fractions of the cell, respectively; this is now shown in Figure 4—figure supplement 2A). We then used liquid chromatography-mass spectrometry (LC-MS) to quantify the amounts of **147** in the cytosol- and ER-containing fractions based on their peak areas by comparison with a standard curve. We found 2.1 ± 0.2 ng of **147** per million cells in the cytosol fraction compared to 5.6 ± 0.9 ng in the ER fraction (this is now shown in Figure 4—figure supplement 2B), corresponding to a ratio of 2.7 ± 0.5. Given that membranous organelles likely constitute a smaller fraction of the total volume of HEK293T cells than the cytosol (even in hepatocytes, membranous organelles are about only 40% of the total cell volume compared to 55% for the cytosol), this likely represents a lower limit for this ratio. This preference is likely due to the hydrophobicity of **147** (estimated logP = 3.35, where “P” is the octanol/water partition coefficient; see Plate et al., 2016), which would lead to its sequestration in the membranes that surround organelles like the ER.

This result suggests that the selectivity of **147** for ER-resident proteins is at least in part the result of its higher concentration in the ER. It is additionally true that the cytochrome P450s that most likely convert **147** to **147-QM** (the quinone methide) reside in the ER membrane. Moreover, as the reviewer notes, the reducing environment of the cytosol would tend to decrease the lifetime of **147-QM** relative to its lifetime in the oxidizing environment of the ER, thereby diminishing the extent of protein modification in the cytosol. We believe that all of these factors probably contribute to the preference of **147** to modify ER-resident proteins.

We have added the following paragraphs to the Results section to reflect these findings:

“To further explore the origins of the selectivity of **147** and its analogs for modifying ER proteins, we sought to determine whether **147** preferentially partitions to the ER. […] The combination of these factors likely drives the preference of **147** to selectively react with ER-resident proteins.”

In addition, we have edited the Discussion section to read as follows:

“A remarkable aspect of this mechanism of pharmacologic ATF6 activation is the preferential covalent modification of proteins localized to the ER by **147**. […] This is especially true in the cytosol, in which **147-QM** likely has a shorter lifetime than in the ER because of the cytosol’s more reducing environment.”

2) The other interesting question is whether **147** treatment elevates PDIA levels through ATF6 activation over time? In other words, does the ATF6 activation lead to a shut-off of the compound activity by increasing the receptor level? In the author's previous work, I think they found that ATF6 activates PDIA4 and PDIA6 (Shoulders et al., 2013). This part of the mechanism is potentially important for the use of **147** in longer term treatment regimes and it might explain the relatively low toxicity.

We agree with the reviewer that the duration of ATF6 induction by **147** is an interesting question. As the reviewer notes, several of the targets of **147** are also induced by **147**-mediated ATF6 activation. However, **147** has been used in experiments in which it was shown that **147**-mediated ATF6 activation could suppress stem cell pluripotency, enhance differentiation, and direct mesodermal cell fate (Kroeger et al., 2018). These results required dosing cells daily with **147** over a period of weeks, suggesting that the induction of PDIs does not lead to a shut-off of the activity of **147**. In unpublished murine studies with dosing at 48 h intervals, we also did not observe attenuation of the activity of **147**.

We have added the following paragraph to the Results subsection “shRNA-depletion of PDIs perturbs **147**-dependent ATF6 activation” to raise this issue and describe our results:

“The notion that **147** acts through modification of PDIs, some of which are in turn upregulated by **147**-mediated ATF6 activation (especially PDIA4, and to a lesser extent PDIA1 and PDIA3) (Plate et al., 2016) raises the question of whether the increased levels of PDIs could diminish the efficacy of **147** as time goes on. […] Thus, while the potential for increased PDI levels to suppress **147**-dependent ATF6 activation offers an interesting mechanism to modulate **147** activity, further studies will be required to elucidate whether such a negative feedback mechanism is operating.”

Reviewer #2:[…] The evidence for apparent selectivity of ER proteins and specifically 6 PDIs by **147** is convincing, and further supported by the genetic evidence using shRNA knockdown that PDIA3, PDIA4, and PDIA5 affect the **147**-dependent regulation of ATF6. The basis for specificity of these oxidized PDIs and perhaps other identified targets on regulation of ATF6 is not examined or much discussed. Presumably the effect of **147** is not due to conformational effects, i.e. causing protein damage and generation of misfolded proteins in the ER, however some of these PDIs are known to interact with calreticulin and other components of the lumen chaperone machinery. This needs to be addressed, either experimentally or minimally by a more thorough discussion as the stress signals for XBP1 and HSF1 are directly linked to increased flux of misfolded proteins. Alternatively, these data might suggest a regulatory process involving electrophile stress similar to Nrf2/Keap1.

The reviewer’s point is well taken. While we have shown by multiplex gene expression profiling, as well as by RNA-seq, that **147** does not broadly induce stress-responsive signaling by the stress sensors XBP1, HSF1 or NRF2, we have also found some compounds with structures similar to **147**, notably including the 2-amino-*p*-cresol warhead, that are substantially less selective. The small molecule **132**, shown in Figure 5—figure supplement 2, activates ATF6, XBP1, and PERK targets to similar extents. Given their structural similarity, it is reasonable to expect that **132** acts by a similar mechanism as **147**; that is, by oxidation to a quinone methide followed by covalent protein modification. Thus, it appears that pushing reactivity too far and/or QM concentration too high can result in a broader stress-responsive signaling pathway activation beyond ATF6. As the reviewer suggests, the PDIs that **147** modifies could interact with other components of the ER proteostasis network in addition to ATF6. The selectivity of **147** likely arises from it having a lower propensity to modify the Cys-containing proteome than **132**, either because **147** is not as reactive toward protein nucleophiles, or because its QM is at lower concentration, or because it is more rapidly quenched.

We have modified the Discussion on as follows to address this issue:

“Collectively, these features combine to make **147** a precision electrophile that preferentially activates the ATF6 signaling pathway without triggering other pathways activated by similar stimuli, such as the IRE1 and PERK UPR signaling pathways, the heat shock response, or the NRF2 oxidative stress response as we have shown previously by multiplex gene expression profiling, as well as by RNA-seq (Plate et al., 2016). […] The selectivity of **147** likely arises from it having a lower propensity to modify the Cys-containing proteome than **132**, either because **147** is not as reactive toward protein nucleophiles, or because its quinone methide is at lower concentration, or because it is more rapidly quenched.”